# Translational control of breast cancer plasticity

Michael Jewer[1,2], Laura Lee[2], Matthew Leibovitch[3], Guihua Zhang[2], Jiahui Liu[2], Scott D. Findlay[1,2], Krista M. Vincent[1,2], Kristofferson Tandoc[3], Dylan Dieters-Castator[1], Daniela F. Quail [4], Indrani Dutta[2], Mackenzie Coatham[2], Zhihua Xu[2], Aakshi Puri[3], Bo-Jhih Guan[5], Maria Hatzoglou[5], Andrea Brumwell[6], James Uniacke [6], Christos Patsis[3], Antonis Koromilas[3], Julia Schueler[7], Gabrielle M. Siegers [2], Ivan Topisirovic [3] & Lynne-Marie Postovit [2,8✉]

Plasticity of neoplasia, whereby cancer cells attain stem-cell-like properties, is required for disease progression and represents a major therapeutic challenge. We report that in breast cancer cells *NANOG*, *SNAIL* and *NODAL* transcripts manifest multiple isoforms characterized by different 5' Untranslated Regions (5'UTRs), whereby translation of a subset of these isoforms is stimulated under hypoxia. The accumulation of the corresponding proteins induces plasticity and "fate-switching" toward stem cell-like phenotypes. Mechanistically, we observe that mTOR inhibitors and chemotherapeutics induce translational activation of a subset of *NANOG*, *SNAIL* and *NODAL* mRNA isoforms akin to hypoxia, engendering stem-cell-like phenotypes. These effects are overcome with drugs that antagonize translational reprogramming caused by eIF2α phosphorylation (e.g. ISRIB), suggesting that the Integrated Stress Response drives breast cancer plasticity. Collectively, our findings reveal a mechanism of induction of plasticity of breast cancer cells and provide a molecular basis for therapeutic strategies aimed at overcoming drug resistance and abrogating metastasis.

[1] Department of Anatomy and Cell Biology, University of Western Ontario, London, ON, Canada. [2] Department of Oncology, University of Alberta, Edmonton, AB, Canada. [3] Lady Davis Institute, Departments of Oncology and Biochemistry, Division of Experimental Medicine, McGill University, Montreal, QC, Canada. [4] Goodman Cancer Center, McGill University, Montreal, QC, Canada. [5] Department of Pharmacology, Case Western Reserve University, Cleveland, OH, USA. [6] Department of Molecular and Cellular Biology, University of Guelph, Guelph, ON, Canada. [7] Charles River Discovery Research Services Germany, Freiburg, Germany. [8] Department of Biomedical and Molecular Sciences, Queen's University, Kingston, ON, Canada. ✉email: l.postovit@queensu.ca

More than 20% of breast cancer patients will die due to therapy resistance and metastasis. Both processes require cancer cells to adapt to stresses like hypoxia or chemotherapy. This adaptability can be mediated by increased plasticity of breast cancer cells enriching for breast cancer cells with stem-cell-like features (BCSC)[1,2].

Hypoxia induces breast cancer cell plasticity and stem-cell-like phenotypes[3–5]. Although this process is poorly understood, reports demonstrate that hypoxia induces the expression of stemness-factors including NANOG, SNAIL, and NODAL[6–8]. Moreover, both stem-cell-like phenotypes and elevated expression of stemness-factors have been linked to metastatic spread and chemoresistance[1]. Hypoxia induces energy conserving reductions in protein synthesis that are mediated by inhibition of the mammalian/mechanistic target of rapamycin complex 1 (mTORC1) and induction of the integrated stress response (ISR) arm of the unfolded protein response[9–11]. mTORC1 inhibition reduces recruitment of mRNA to the ribosome via 4E-BP-dependent inhibition of eIF4F complex assembly (eIF4E:eIF4G:eIF4A)[12]. EIF2α phosphorylation, a hallmark of the ISR, decreases ternary complex (eIF2:tRNAiMet:GTP) recycling, thereby limiting initiator tRNA delivery[13]. Despite decreased protein synthesis, some mRNAs are preferentially translated, facilitated by features in their 5′ untranslated region (5′UTR), such as upstream open-reading frames (uORFs)[14]. Together, these pathways drive an adaptive stress response, conserving energy while enabling the expression of pro-survival factors, including ATF4[15]. Emerging data suggest that mTOR inhibition and eIF2α phosphorylation promote stem-cell-like phenotypes. For instance, mTOR inhibition promotes self-renewal in neural stem cells by activating 4E-BPs[16,17] and eIF2α phosphorylation is essential for the maintenance of self-renewal in satellite cells and hESCs[18,19]. These studies show that translational reprograming may regulate stem-cell-associated plasticity, but the role of this process in the acquisition of BCSC phenotypes remains elusive.

Herein, we demonstrate that breast cancer cells possess multiple transcript isoforms of NANOG, SNAIL, and NODAL that differ in their 5′UTRs, some of which show preferential translation in hypoxia facilitating increased protein expression. This translationally induced stem cell program leads to the acquisition of BCSC phenotypes. Like hypoxia, mTOR inhibition and chemotherapeutics also induce plasticity via translational reprogramming. Finally, we demonstrate that inhibiting the ISR with the *I*ntegrated *S*tress *R*esponse *I*nhi*B*itor (ISRIB) impedes acquisition of stem-cell-like phenotypes and therapy resistance of breast cancer cells.

## Results

**Hypoxia-induced NODAL mediates breast cancer plasticity.** Stem cell-associated proteins, including NANOG, SNAIL, and NODAL, have been shown to induce epithelial to mesenchymal transition (EMT) and are associated with poor outcomes in breast cancer patients[20–23]. NANOG and SNAIL can also induce BCSCs[20,21,24]; but notwithstanding a single report[25], NODAL's role in the induction of breast cancer stem-like phenotypes is not well established. Since NODAL may co-operate with factors such as SNAIL and NANOG to support plasticity and induction of stem-like properties[26], we investigated whether NODAL induces BCSC phenotypes. We employed widely used functional readouts for BCSCs, such as single-cell sphere formation assays[27,28] and flow cytometric monitoring of CD44high/CD24low cells[2,29] (Gating in Supplementary Fig. 7). Recombinant human NODAL (rhNODAL) increased sphere formation in luminal-like T47D and MCF7 cells (Fig. 1a; Supplementary Fig. 1a), and enriched the population of CD44high/CD24low cells in MCF7, T47D, and

triple-negative SUM149 cells (Supplementary Fig. 1b–d). NODAL was up-regulated in BCSC-enriched 3D cultures (Fig. 1b), and NODAL depletion or NODAL signaling inhibition with activin receptor-like kinase (ALK) inhibitor[30] (SB431542, 1 µM) reduced sphere formation by MDA-MB-231 cells (Fig. 1c,d). These data corroborated that NODAL promotes BCSC phenotypes in cell lines representing luminal-like and triple-negative breast cancer subtypes.

Hypoxia is thought to increase the plasticity of cancer cells, enriching BCSC phenotypes. Therefore, we investigated the role of NODAL in hypoxia-induced plasticity. T47D and SUM149 cells were maintained under hypoxia for 24 h. We observed hypoxia-associated increases in tumorsphere formation, as well as an increase in CD44high/CD24low cells with no reduction in cell viability (Fig. 1e, f; Supplementary Fig. 1e), suggesting that hypoxia induces BCSCs. Moreover, RNA sequencing of T47D cells cultured for 48 h in 20% or 1% $O_2$ demonstrated that hypoxia also entices an EMT signature (Fig. 1g; Supplementary Table 1). Finally, SB431542 treatment abrogated hypoxia-induced sphere formation in T47D and SUM149 cells, corroborating that NODAL facilitates plasticity in response to hypoxia in luminal and triple-negative breast cancer cell lines (Fig. 1h; Supplementary Fig. 1f).

**Translation up-regulates NODAL, NANOG, and SNAIL in hypoxia.** To investigate how hypoxia can induce stem cell-associated proteins, like NODAL, in breast cancer cells, we examined NODAL protein and mRNA expression in T47D cells exposed to hypoxia for 0–24 h. NODAL protein levels increased 2-to-3-fold between 6 and 24 h of hypoxia (1% $O_2$) whereas NODAL mRNA levels were reduced at 3 h and partially recovered by 24 h (Fig. 1i; Supplementary Fig. 1g). In SUM149 cells, a similar discordance between SNAIL mRNA and protein levels was observed (Fig. 1j). In T47D cells, increases in SNAIL and NANOG protein levels appeared to exceed the up-regulation of their transcripts (Supplementary Fig. 1h). These findings strongly suggest that NODAL, SNAIL, and NANOG protein expression is regulated translationally in hypoxia.

To evaluate translation, we employed polysome profiling, which separates efficiently versus inefficiently translated mRNAs by sucrose gradient ultracentrifugation[31]. A 24-h hypoxia treatment caused a 40–90% reduction in global translation in T47D, MCF7, and H9 cells (Fig. 1k, Supplementary Fig. 1i, j) as reported in other systems[11,32]. Using digital droplet RT-PCR (ddPCR) comparing total and efficiently translated mRNA fractions (associated with >3 ribosomes), we assessed polysomal distribution of known translationally suppressed or induced mRNAs under hypoxia[14]. Expectedly, in T47D cells hypoxia reduced translation of 5′ terminal oligopyrimidine (TOP) containing eukaryotic elongation factor 2 (EEF2) and ribosomal protein, large, P0 (RPLPO) mRNAs (Fig. 1l) while enhancing the translation of activating transcription factor 4 (ATF4) and vascular endothelial growth factor (VEGF) mRNAs (Fig. 1l). In hESCs, the translation of NANOG, NODAL, and SNAIL mRNAs was either sustained or increased under hypoxia, similar to VEGF and ATF4 and in contrast to RPLPO (Fig. 1m). Stresses like hypoxia cause adaptive translational reprogramming via modulating mTOR and ISR signaling[33–36]. Immunoblotting confirmed that in T47D cells, hypoxia reduces mTORC1 activity—illustrated by decreased phosphorylation of eIF4E-binding protein 1 (4E-BP1) and ribosomal protein S6 (rpS6) (1% $O_2$; 24 h), while inducing ISR as evidenced by increased eIF2α phosphorylation (Fig. 1n, Supplementary Fig. 1k). VEGF protein was concurrently up-regulated (Fig. 1n, Supplementary Fig. 1k). Similar results, confirming hypoxia induces translational

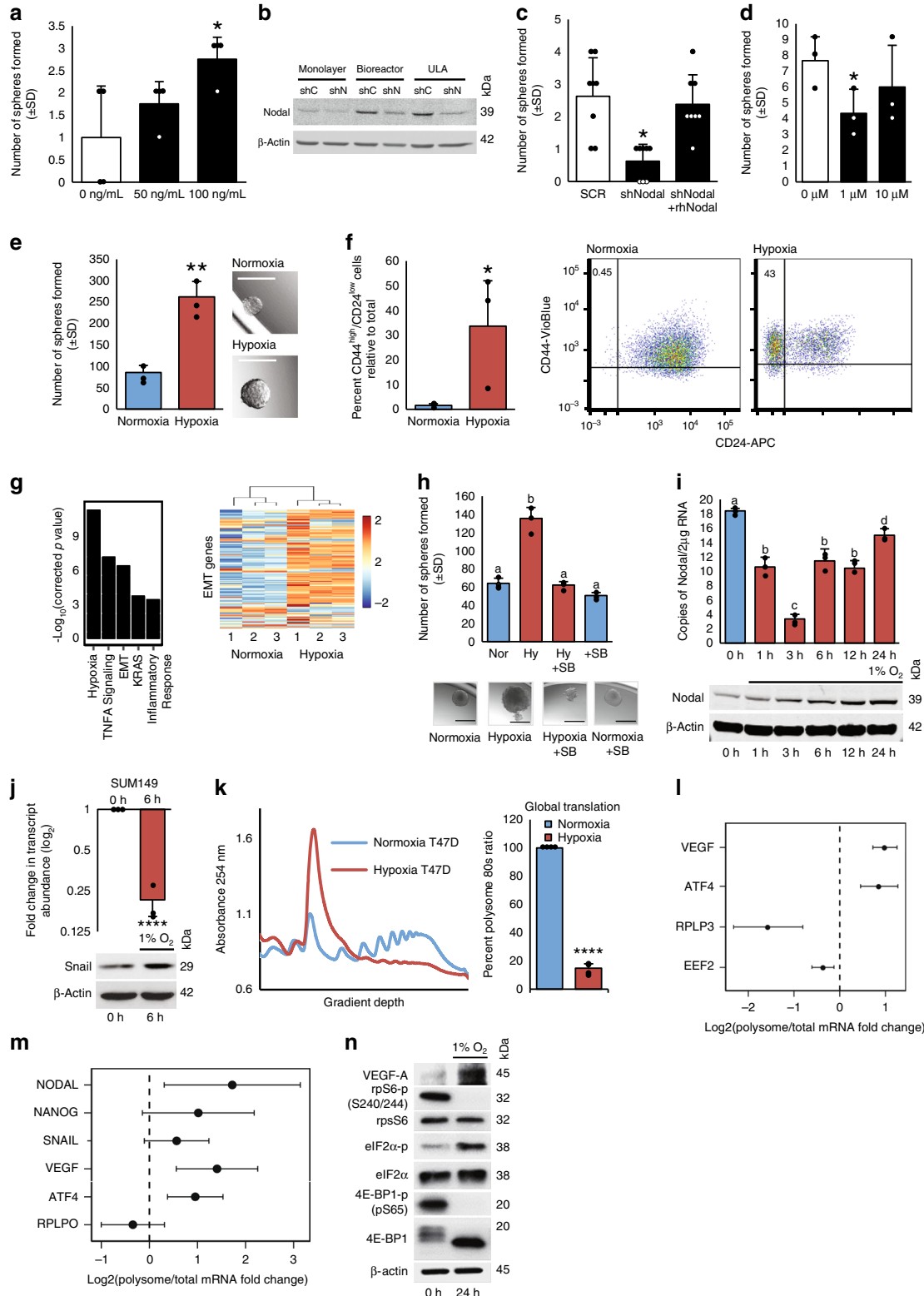

reprogramming by inhibiting mTORC1, and eIF2α phosphorylation was observed in MCF7 and H9-hESC cells, wherein electrophoretic shifts in total 4E-BP1 indicate a reduction in phosphorylation, coinciding with increased eIF2α phosphorylation (Supplementary Fig. 1l). These results suggest that translation of the stemness-factor-encoding mRNAs NANOG, NODAL, and SNAIL is up-regulated during hypoxia similar to the ISR-induced translation of ATF4 or cap-independently translated VEGF transcripts.

**Isoform-specific 5′UTRs enable translation in hypoxia**. To determine the mechanisms responsible for maintaining the translation of SNAIL, NODAL, and NANOG mRNAs under hypoxia we used RefSeq and publicly available CAGE data, in combination with 5′RACE to examine their 5′UTRs, as translational efficiency is largely determined by 5′UTR features[14]. We discovered that the NODAL, SNAIL, and NANOG genes contain multiple transcriptional start sites (TSSs), which result in mRNA isoforms that differ in their 5′ UTRs, but not in their coding

**Fig. 1 Hypoxia-induced translational reprogramming promotes BCSC phenotypes. a** Mean number of spheres formed from T47D cells treated with rhNODAL (0, 50, 100 ng/mL) relative to untreated cells ($n = 3$). **b** NODAL immunoblots of MDA-MB-231 cells: 2D monolayer, 3D bioreactor, and tumorspheres grown in Ultra Low Attachment (ULA) plates. Cells expressed shRNA agianst NODAL (shN) or a control (shC) and β-actin is a loading control. Mean spheres formed from MDA-MB-231 **c** expressing NODAL shRNA with or without rhNODAL (100 ng/mL) or **d** exposed to SB431542 (1, 10 μM) ($n = 3$). **e** Mean spheres from 960 T47D cells pre-exposed to 24 h hypoxia or normoxia ($n = 3$) with representative images. Micron bars = 250 μm. **f** Mean percentage of CD44$^{high}$/CD24$^{low}$ T47D cells following 24 h hypoxia or normoxia ($n = 3$). Representative scatterplots defining CD44 and CD24 subpopulations. **g** RNAseq gene set enrichment analysis from T47D cells cultured 48 h in normoxia or hypoxia. Heatmap of EMT genes demonstrating hypoxic up-regulation ($n = 3$). **h** Mean spheres formed from 960 T47D cells pre-exposed to 24 h hypoxia or normoxia ± SB431542 (10 μM) with representative images ($n = 3$). Micron bars = 500 μm. **i** Mean *NODAL* transcript copy number qRT-PCR vs. known standards and protein levels (immunoblot) in hypoxia-treated (0–24 h) T47D cells ($n = 3$). **j** *SNAIL* transcript mean log$_2$-fold change (qRT-PCR) and protein levels (immunoblot) in hypoxia-treated SUM149 cells (0, 6 h) ($n = 3$). **k** Polysome profile and global translation in hypoxia versus normoxia in T47D cells. Mean percent polysome-associated mRNA (>3 ribosomes). **l** Mean log$_2$-fold change of the ratio of polysome-associated *EEF2, RPLPO, VEGF,* and *ATF4* mRNA levels in T47D cells used in **k** and **m** polysome-associated *RPLPO, VEGF, ATF4, NANOG, SNAIL,* and *NODAL* mRNA levels in H9 hESC cultured for 24 h in 1 versus 20% O$_2$ ($n = 3$). **n** 4E-BP, rpS6, and eIF2α phosphorylation immunoblots from T47D cells used in **k**. 4E-BP1, rpS6, eIF2α, and β-actin-loading controls and hypoxia-positive control VEGF. Hypoxia denoted in red and normoxia blue. Data represents independent experiments. Error bars indicate mean ± SD. Two-sided *t*-test for paired samples. The asterisks denote *p*-values < *0.05, **0.01, ****0.0001. Multiple comparisons tested by ANOVA. The same letters indicate relationships with a $p \geq 0.05$. Different letters indicate statistical differences ($p < 0.05$).

---

sequences (Fig. 2a–c). In the *NANOG* locus, we validated a previously described 350 nucleotides (nt) 5′UTR[37] as well as an alternative 291 nt 5′UTR (Fig. 2a). We observed two TSSs in the *SNAIL* locus: one yielding a 417 nt 5′UTR and another that generates a 85 nt 5′UTR (Fig. 2b). In the *NODAL* locus, there were four 5′UTRs comprised of 14, 42 and 298 and 416 nt (Fig. 2c). We hypothesized that the availability of multiple mRNA isoforms with identical ORFs but divergent 5′UTRs might facilitate the translation of NANOG, SNAIL, and NODAL in both stressed (e.g. hypoxia) and unstressed conditions. Some of these 5′UTRs enable selective translation whereas others are more functional when translation is unperturbed. This contrasts with single-isoform-mRNAs, such as *ATF4* that are translationally repressed in non-stressed conditions and preferentially translated during stress. To investigate if *NODAL, SNAIL,* and *NANOG* isoforms are differentially translated under hypoxia, MCF7 cells cultured in 20% or 1% O$_2$ for 24 h were fractionated to obtain mRNAs associated with monosomes, light and heavy polysomes (Supplementary Fig. 2a). The percentage of transcript associated with each fraction was then determined using isoform-specific qRT-PCR. As positive controls for transcripts that are selectively translated during ISR and hypoxia[38], we confirmed that *ATF4* and *VEGFA* transcripts were shifted towards heavy polysomes in cells cultured in hypoxia versus normoxia (Supplementary Fig. 2b and c). In contrast, β-*ACTIN* mRNA, was translationally suppressed under hypoxia (Supplementary Fig. 2d). The 5′UTR mRNA isoforms showed differential responses. The translation of the *NANOG* 350 mRNA isoform was maintained in hypoxia, but the *NANOG* 291 mRNA isoform was translationally induced in hypoxia relative to normoxia (Fig. 2d). In turn, the longer *SNAIL* mRNA isoform was less efficiently translated in hypoxia compared to the shorter 5′UTR isoform (Fig. 2e). Finally, translational efficiency of the *NODAL* 298 and 416 5′UTR mRNAs, was increased under hypoxia to a greater extent than the shorter (42 and 14 nt) isoforms (Fig. 2f). These findings suggest a model of translational regulation where mRNA isoforms with different 5′UTR features enable translation and commensurate accumulation of NANOG, SNAIL, and NODAL proteins under hypoxia.

To further test this model, we cloned each 5′UTR of NANOG, SNAIL, and NODAL into a p5′UTR firefly construct[39], transfected cells, exposed them to 20% or 1% O$_2$ and then measured luciferase activity. These reporter assays served as surrogates for endogenous isoform expression and enabled the analysis of each 5′UTR without potential confounding elements in the coding region or 3′UTR. In accordance with the polysome-profiling data, both 5′UTRs of NANOG facilitated translation in

hypoxia, whereas the shorter SNAIL 5′UTR facilitated translation significantly more than the longer SNAIL 5′UTR (Fig. 2g, h). We also found that the 416 and 298 nt NODAL 5′UTRs increased translational efficiency in hypoxia more than the 42 nt or 14 nt 5′UTRs (Fig. 2i). The 298 and 416 nt NODAL 5′UTRs contain one and four putative uORFs, respectively. Due to the complex organization and splicing of the 416nt 5′UTR we examined the functionality of the single uORF in NODAL 298 (Supplementary Fig. 2e). *ATF4* is well-established as a mRNA that is translationally activated during ISR in a uORF-dependent manner[39–41]. Mutation of the start codon of uORF1 abrogates the ISR-mediated override of the inhibitory uORF2 and allows for preferential translation of the main ORF[39–41]. In order to analyze the translational regulation conferred by the NODAL 298 uORF we transfected cells with p5′UTR firefly constructs harboring a wildtype or uORF mutant NODAL 298 5′UTR, as well as wildtype or uORF1 mutant ATF4 5′UTR[39]. As expected, luciferase activity in the cells expressing the ATF4 5′UTR construct was enhanced by both hypoxia and thapsigargin (TG, 100 nM, as a positive control for ISR), and this was abrogated by the uORF1 mutation (Fig. 2j). Similar to the ATF4 5′UTR, hypoxia enhanced translation from the NODAL 298 5′UTR, and this effect was attenuated when the uORF was mutated (Fig. 2k). Notably, however, the function of the uORF in NODAL 298 differed from that in ATF4 as it possesses an apparent pro-translation function in unstressed normoxic conditions. NODAL expression under hypoxia is achieved through selective translation of isoforms with longer uORF containing 5′UTRs. This mode of translational regulation is distinct from ATF4's translational activation under ISR. ATF4 has a single isoform that is activated by delayed re-initiation, whereas NODAL differentially utilizes specific uORFs harboring isoforms for efficient translation. In contrast to NODAL, shorter 5′UTR isoforms of *SNAIL* and *NANOG*, seemingly devoid of putative uORFs, are translated more efficiently under hypoxia than their longer counterparts. This difference may be explained by the lower energy requirements needed to translate shorter and less complex 5′UTRs. These results indicate that isoform-specific translation of NANOG, SNAIL, and NODAL mRNAs is responsible for increasing their expression in hypoxia, notwithstanding differences in the mechanisms by which specific 5′UTR features regulate their translation.

**mTOR inhibition differentially selects for 5′UTR utilization.** Our results suggest that hypoxia may increase BCSC phenotypes through selective translation driving the synthesis of NANOG,

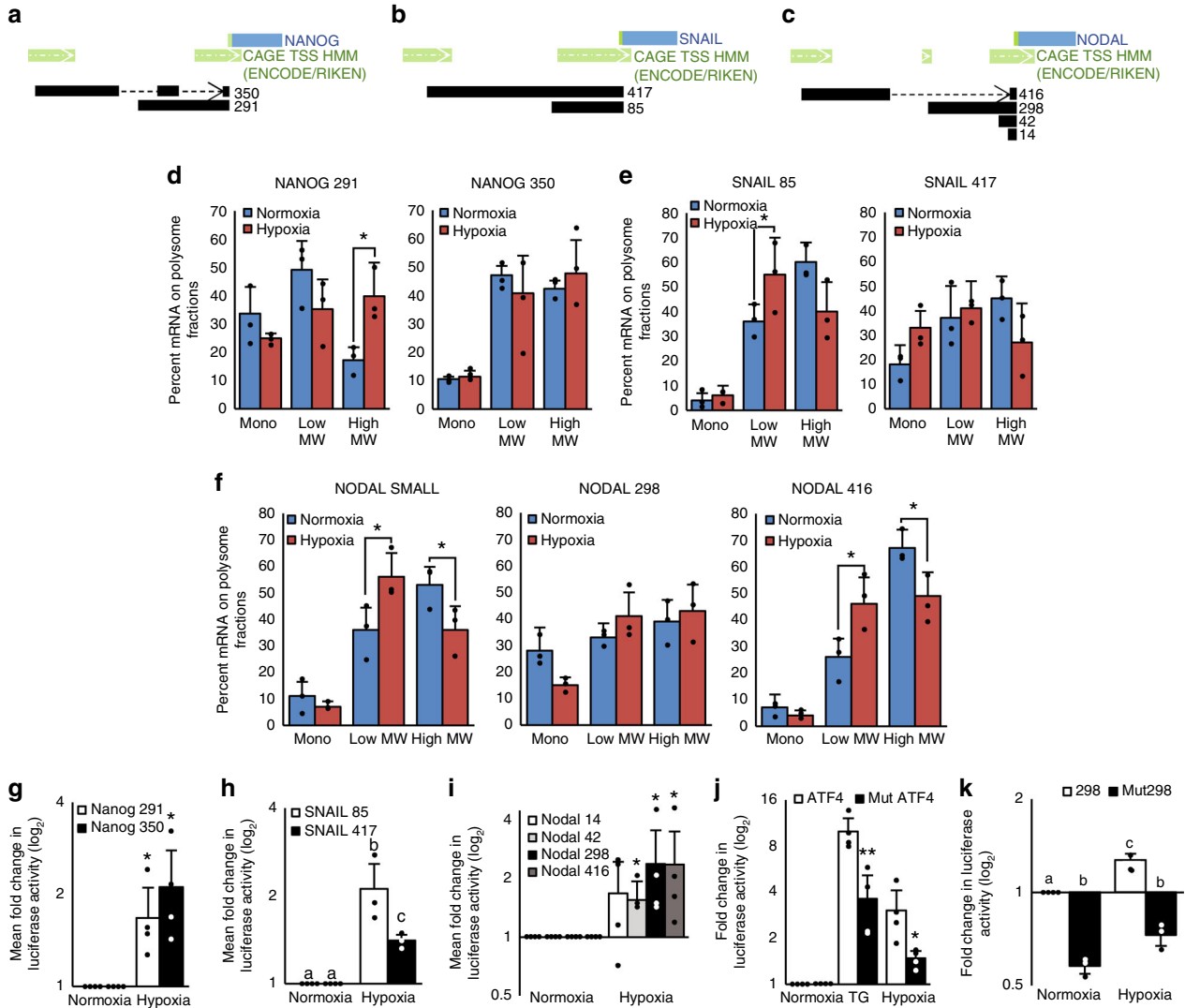

**Fig. 2 Isoform-specific selective translation of BCSC-associated transcripts in hypoxia. a–c** Diagrams of the **a** NANOG, **b** SNAIL, and **c** NODAL gene loci showing multiple transcriptional start sites (dark green with white arrows) leading to multiple 5′UTR sequences (black). Dotted lines represent splicing events. The blue rectangles represent the ORFs and the bright green bar represents the translational start site. **d–f** mRNA isoforms of **d** NANOG, **e** SNAIL, and **f** NODAL associated with mononosomes, low MW polysomes and high MW polysomes extracted from MCF7 cells cultured for 24 h in hypoxia or normoxia. Bars represent the mean percent of transcript associated with each fraction in each condition ± SD ($n = 3$, NODAL $n = 4$). Lines and asterisk (*) indicate significant differences between conditions ($p < 0.05$). **g–i** Luciferase activity in cells transfected with p5′UTR firefly constructs harboring the various 5′UTRs of **g** NANOG ($n = 4$), **h** SNAIL ($n = 3$), or **i** NODAL ($n = 4$) and then exposed for 6 h to normoxia or hypoxia. Bars represent mean luciferase activity in hypoxia relative to normoxia ± SD. Different letters are statistically different in **h** and values indicated by an asterisk (*) in **g** and **i** are statistically different from control values but not from each other ($p < 0.05$). **j** Luciferase activity in cells transfected with p5′UTR firefly constructs harboring either the wild type or uORF1 mutated ATF4 5′UTR and then exposed for 6 h to normoxia, thapsigargin (TG; 0.1 µM), or hypoxia. Bars represent mean luciferase activity relative to cells exposed to normoxia ± SD. Values indicated by an asterisk (*) are statistically different from wild type 5′UTR ($p < 0.02$, $n = 3$). **k** Luciferase activity in cells transfected with p5′UTR firefly constructs harboring either the wild type or uORF mutated NODAL 298 5′UTR and then exposed for 6 h to normoxia or hypoxia. Bars represent mean luciferase activity relative to cells exposed to normoxia ± SD ($n = 3$). Error bars indicate mean ± SD. Two-sided $t$-test for paired samples. The asterisks denote $p$-values < *0.05, **0.01. Multiple comparisons tested by ANOVA. The same letters indicate relationships with a $p ≥ 0.05$. Different letters indicate statistical differences ($p < 0.05$).

SNAIL, and NODAL proteins. Mechanistically, selective translation occurs because NANOG, SNAIL, and NODAL have multiple 5′UTRs, some of which are translated during global protein synthesis inhibition. Accordingly, we hypothesized that mTOR inhibition (which inhibits protein synthesis and induces eIF2α phosphorylation) might induce BCSCs concomitant with the selective translation of specific 5′UTR isoforms of NODAL, NANOG, and SNAIL as demonstrated in hypoxia (Fig. 3a). To this end, we investigated the effect of the active-site mTOR inhibitor INK128 and ISRIB on global and mRNA isoform

(NODAL, SNAIL, and NANOG) specific translation. MCF7 cells cultured in vehicle, ISRIB (10 µM), INK128 (INK; 20 nM) or INK128 + ISRIB for 24 h, were fractionated to obtain mRNAs associated with monosomes, light and heavy polysomes. INK128 reduced global protein synthesis concomitant with mTORC1 suppression and ISR activation illustrated by reduced 4E-BP1 and rpS6 phosphorylation, and elevated eIF2α phosphorylation, respectively (Fig. 3b; Supplementary Fig. 3a–d). ISRIB counteracts the phospho-eIF2α-dependent inhibition of TC recycling by bolstering eIF2B GEF activity[42,43]; however,

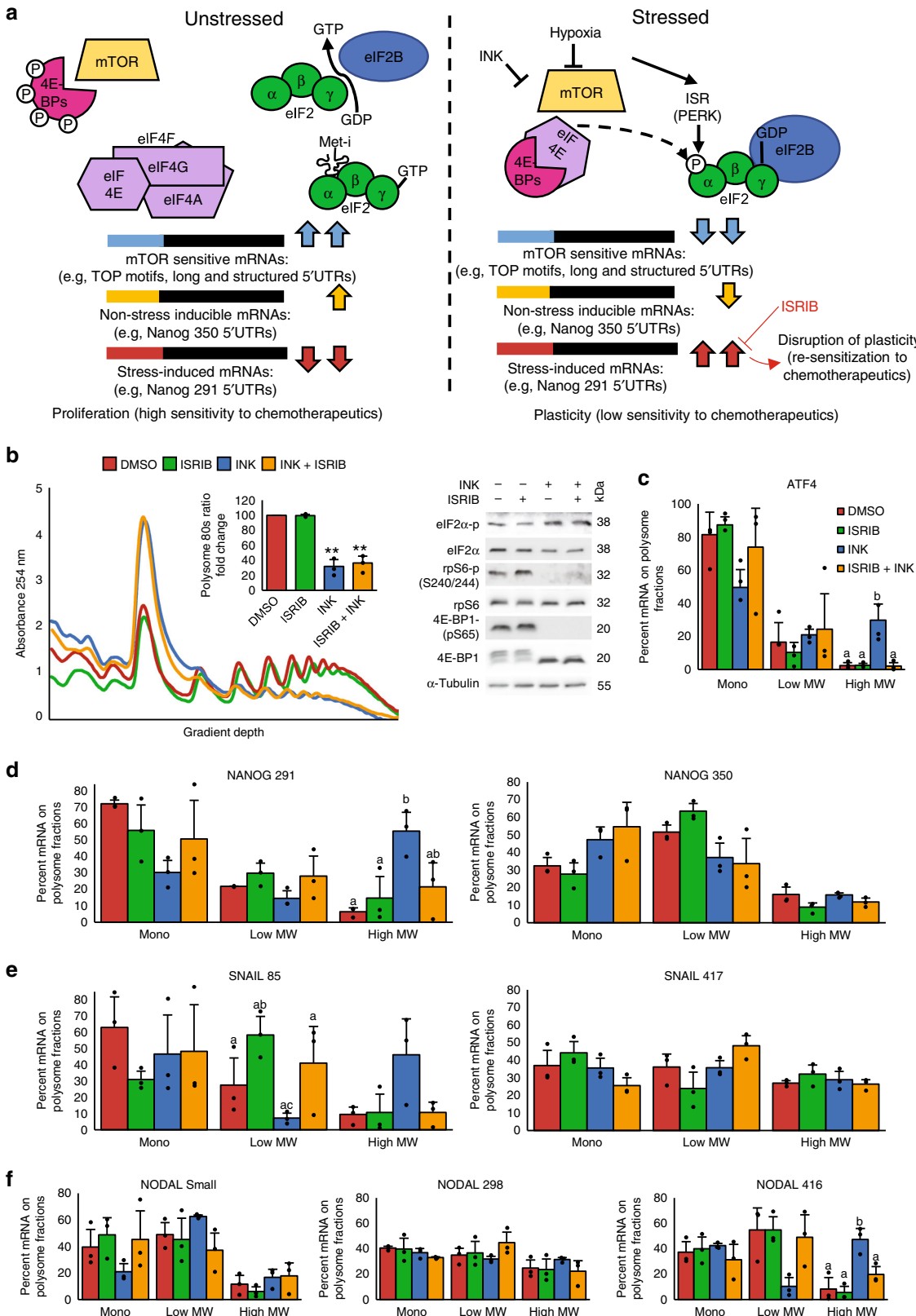

cotreatment with ISRIB did not mitigate the global reduction in translation in response to INK128 (Fig. 3b). To identify if mRNA isoform utilization was altered, the percentage of transcript associated with each fraction was then determined using isoform-specific qRT-PCR. As a control for ISR-driven selective translation, we confirmed that *ATF4* mRNA was shifted towards heavy

polysomes in cells cultured in INK128 versus controls. ISRIB cotreatment reversed this translational upregulation (Fig. 3c), confirming that ISRIB targets the ISR-mediated effect of mTOR inhibition. Similar to the effects of hypoxia, INK128 induced a shift of *NANOG* 291 mRNA to heavy polysomes while translational efficiency of the *NANOG* 350 mRNA was slightly reduced

**Fig. 3 Isoform-specific translation of BCSC transcripts during mTOR inhibition. a** Diagram of a working model wherein hypoxia causes a reduction in mTOR activity and the activation of the integrated stress response (ISR), signified by eIF2α phosphorylation. These changes in the translational machinery lead to diminished global protein synthesis and selective translation of a subset of mRNA isoforms of NANOG, SNAIL, and NODAL that contain specific 5′ UTR features, including uORFs. The experiments presented below are designed to test whether mTOR inhibition (treatment with INK128) can replicate the phenomena observed during hypoxia and whether ISRIB can mitigate these effects. **b** Polysome profiles and immunoblot analysis of lysates from MCF7 breast cancer cells exposed to vehicle (DMSO), MLN0128/INK128 (INK; 20 nM), ISRIB (10 nM), or INK + ISRIB for 24 h show that INK reduces global translation (INK and ISRIB + INK $p < 0.01$) concomitant with reduced 4E-BP1 and rpS6 phosphorylation and increased eIF2α phosphorylation. These readouts are unaffected by ISRIB. Total 4E-BP, rpS6, and eIF2α levels were unchanged. β-actin is used as a loading control. **c** ATF4 mRNA and **d–f** mRNA isoforms of **f** NANOG, **e** SNAIL, and **f** NODAL associated with monosomes, low MW polysomes and high MW polysomes from MCF7 cell lysates fractionated in **b**. Bars represent the mean percent of transcript associated with each fraction in each condition ± SD ($n = 3$). Error bars indicate mean ± SD. Multiple comparisons tested by ANOVA. The asterisks denote $p$-values < *0.05, **0.01 compared to control treatment. Multiple comparisons tested by ANOVA. The same letters indicate relationships with a $p \geq 0.05$. Different letters indicate statistical differences ($p < 0.05$).

(Fig. 3d). Likewise, the shorter 5′UTR SNAIL isoform was shifted to heavy polysomes while the translation of the longer 5′UTR SNAIL mRNA isoform was maintained (Fig. 3e). Finally, translational efficiency of the NODAL 298 and 416 5′UTR mRNAs was maintained or increased under mTOR inhibition (Fig. 3f). Like in hypoxia, the availability of alternative 5′UTRs enables the translation of NANOG, SNAIL, and NODAL mRNAs when mTOR is inhibited. Importantly, we also determined that ISRIB mitigates the selective translation of these mRNA isoforms (Fig. 3d–f). Collectively, these results suggest mTOR inhibition promotes translation of mRNAs encoding stem cell factors, such as NANOG, SNAIL, and NODAL via ISR-dependent selective translation of specific 5′UTR isoforms.

**mTOR inhibition induces BCSC phenotypes.** Clinical trials of mTOR inhibitors in breast cancer patients have shown limited survival benefits[44]. Hypoxia reduces mTOR activity (Fig. 1n) and induces plasticity and stem-cell-like phenotypes in breast cancer cells (Fig. 1e–g). Moreover, like hypoxia, mTOR inhibition allows selective translation of NANOG, SNAIL, and NODAL mRNA isoforms (Fig. 3). Accordingly, we investigated the role of mTOR in the induction of stem-cell-like phenotypes. T47D and MCF7 cells treated for 1–24 h with the mTOR inhibitor INK128 (INK; 20 nM) up-regulated NODAL protein levels over two-fold between 6 and 12 h, as compared to control cells (Fig. 4a; Supplementary Fig. 4a). In T47D, MCF7, and SUM149 cells, INK128 increased sphere-forming frequency and anchorage-independent growth to the same extent as rhNODAL (100 ng/mL), which was used as a positive control (Fig. 4b, c; Supplementary Fig. 4b–e). Cotreatment with SB431542 (10 µM) attenuated INK128-induced sphere formation and anchorage-independent growth suggesting that NODAL signaling mediates INK128's effects on BCSC phenotypes, independent of breast cancer subtype. Consistently, the percentage of CD44high/CD24low cells in T47D, MCF7, and SUM149 cell lines was increased by INK128. This increase was blocked with SB431542 (Fig. 4d; Supplementary Fig. 4f, g). BCSCs are also characterized by the expression of stem cell markers, such as NANOG and OCT4 and have typically undergone EMT[1]. INK128 induced a stem-cell-like EMT signature in T47D and SUM149 breast cancer cells relative to vehicle-treated control cells (Fig. 4e, f; Supplementary Fig. 4h, i). The maintenance of this effect after a 24 h recovery period, suggests that INK128 may induce reprogramming which engenders plasticity and sustains stem-cell-like properties in breast cancer cells. Considering the close link between stem-cell-like properties and metastasis, we treated breast cancer cells with INK128 for 24 h, injected them into the lungs of mice through the tail vein, and then quantified metastases 8 weeks later using HLA staining. This model served as a form of limiting dilution assay, such that tumor-initiating frequency could be determined (as the number of metastases formed per number of cells injected) without confounding

variables (such as alterations in angiogenesis needed for larger tumor growth) that may affect the interpretation of subcutaneous limiting dilution assays. Using this method, we determined that INK128 pretreatment increases the ability of SUM149 cells to initiate metastatic tumor formation in the lung (Fig. 4g).

To determine whether mTOR inhibition increases populations of stem-like cells in established heterogeneous tumors, we treated mice harboring a triple-negative breast cancer patient-derived xenograft (PDX401) with INK128 (30 mg/kg, every second day for 2 weeks starting when tumors reached 5 mm in diameter). Immunohistochemical (IHC) analyses confirmed that phospho-4E-BP1 levels were inversely correlated with hypoxia [delineated by carbonic anhydrase 9 (CA9)] and were downregulated in mice treated with INK128 (Supplementary Fig. 4j). Tumor growth was not altered by INK128 administration and actually increased upon treatment withdrawal (Fig. 4h). Moreover, cancer cells dissociated from tumors grown in INK128-treated animals had higher BCSC frequencies than those treated with vehicle (Fig. 4i). Collectively, these data provide strong evidence that mTOR inhibition induces breast cancer cell plasticity and stemness in vivo.

To determine whether modulating the translational machinery downstream of mTOR regulates NODAL expression, we altered the expression of 4E-BP1- mTOR's major mediator of translational repression[45]—in T47D and MCF7 cells. 4E-BP1 overexpression increased NODAL levels in both cell lines (Fig. 4j; Supplementary Fig. 4k). NODAL levels were relatively unaffected by 4E-BP1 knock-down, likely due to low basal levels of NODAL and redundancies associated with the expression of 4E-BP2 and 4E-BP3. Importantly, 4E-BP1 overexpression also increased sphere formation (Fig. 4k; Supplementary Fig. 4l). Previous studies have shown high 4E-BP1 levels in breast tumors and that 4E-BP1 may participate in hypoxia-associated translational reprogramming[38]. Accordingly, analysis of RNA sequencing data from 1100 breast cancer patients in the TCGA demonstrated that high levels of 4E-BP1 in the primary tumor predict poor survival in the first 5 years post-diagnosis (Fig. 4l).

**The ISR mediates BCSC induction in response to mTOR inhibition.** Hypoxia[11,41] and acute mTOR inhibition[46,47] have been shown to enhance eIF2α phosphorylation. We observed that hypoxia (Fig. 1n; Supplementary Fig. 1l) and mTOR inhibition increase eIF2α phosphorylation after 24 h (Fig. 3b). Shorter duration of INK128 treatments (but not vehicle control) also cause a ~2-fold increase in eIF2α phosphorylation in T47D and SUM149 cells (Fig. 5a; Supplementary Fig. 5a,b). ISR activation (marked by eIF2α phosphorylation) was required for the efficient translation of stem cell proteins in the presence of INK128 (Fig. 3), hence we sought to determine whether the ISR may generally induce stem cell proteins like NODAL and if eIF2α phosphorylation is required for hypoxia-induced stem cell-like phenotypes.

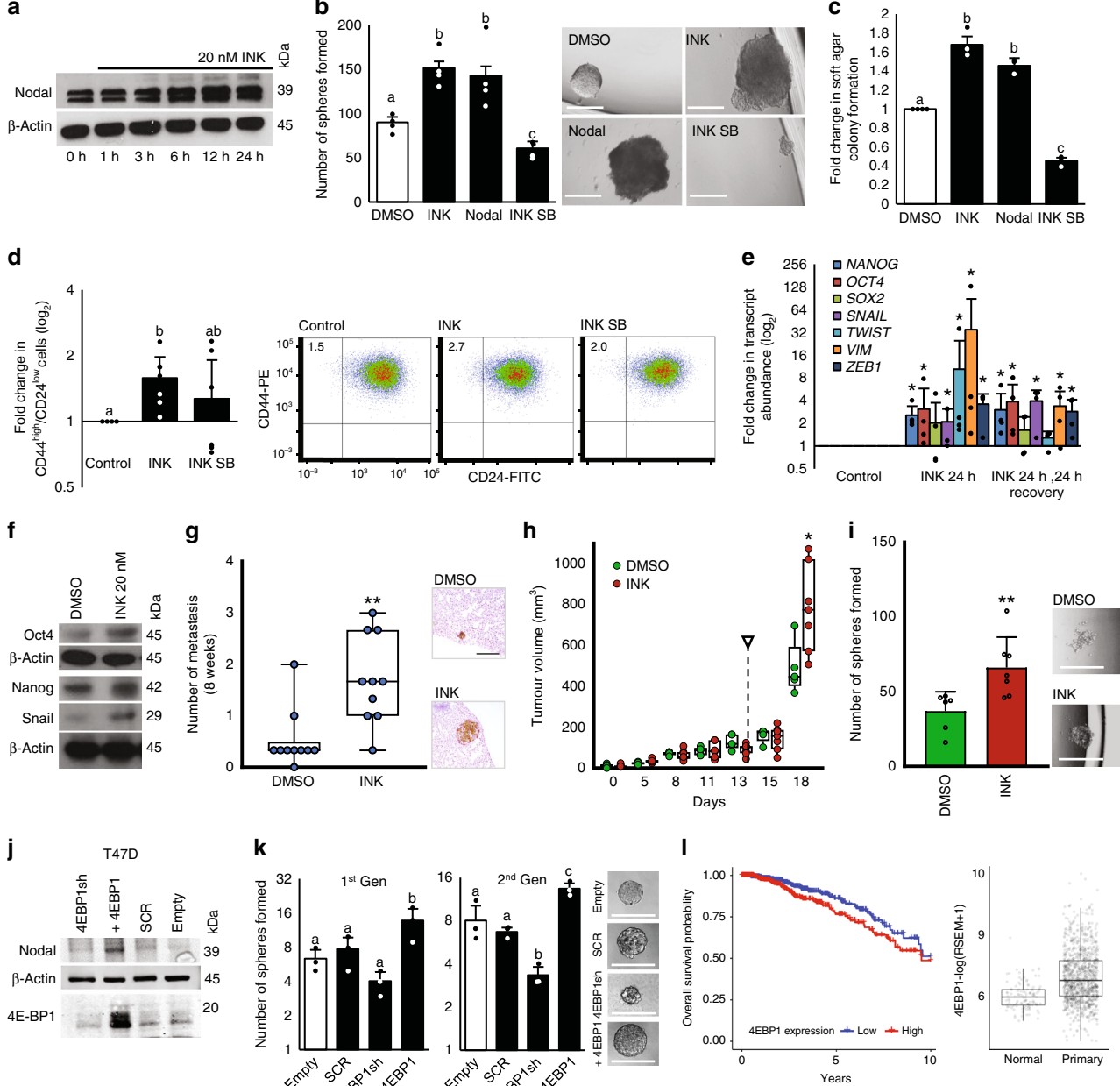

**Fig. 4 MTOR inhibition induces breast cancer plasticity. a** NODAL immunoblot from 0 to 24 h INK 20 nM treated T47D cells ($n = 3$). **b** Mean sphere counts from 960 T47D cells pre-exposed to DMSO, INK (20 nM), rhNODAL (100 ng/mL), or INK + SB431542 (10 μM) for 24 h ($n = 4$) with representative images. Micron bars = 250 μm. **c** Mean fold change in colony number from T47D cells cultured as in **b** ($n = 3$). **d** Mean percentage of CD44high/CD24low T47D cells treated with DMSO, INK, or INK + SB431542 ($n = 6$) with representative scatterplots defining CD44 and CD24 subpopulations. **e** Mean log$_2$-fold change relative to DMSO qRT-PCR of *NANOG, SOX2, OCT4, TWIST, ZEB1, SNAIL, VIM* transcripts from DMSO or INK (20 nM) for 24 or 24 h INK then 24 h media treated T47D cells ($n = 3$). **f** Immunoblot analyses of NANOG, OCT4, and SNAIL in 24 h DMSO or INK (20 nM) treated T47D cells. **g** Mouse lung colonies 8 weeks following tail vein injection of pretreated SUM149 cells (24 h with DMSO or INK (20 nM)) ($n = 10$) with representative lung colony images (brown). **h** Mean PDX401 tumor volumes in mice receiving DMSO or INK (30 mg/kg) every second day for two weeks. Arrow indicates treatment cessation ($n = 8$). **i** Mean sphere counts from 96,000 PDX401 cells from endpoint tumors from **h** (DMSO $n = 6$, INK $n = 7$) with representative images. Micron bars = 500 μm. **j** NODAL and 4E-BP1 immunoblot of T47D cells transfected with 4E-BP1 shRNA, shRNA scrambled (SCR) control, 4E-BP1 ORF, or empty vector. **k** Mean first and second generation sphere counts from 96 T47D cells described in **j** ($n = 3$) with representative images. Micron bars = 250 μm. **l** Kaplan–Meier plot: 4E-BP1 RNA-Seq expression correlates with survival in 1100 breast cancer patient samples (5.4 years, $p = 0.0187$). 4E-BP1 RNA-Seq transcript abundance in tumor versus healthy tissue (normal, $n = 113$; tumor, $n = 1062$; $p = 2.2 \times 10^{-16}$). Data represents independent experiments. Error bars indicate mean ± SD. Box and whisker plot represents median, IQR, whiskers extend to maximum and minimum. Two-sided *t*-test for paired samples. The asterisks denote *p*-values < *0.05, **0.01. Multiple comparisons tested by ANOVA. The same letters indicate relationships with a $p \geq 0.05$. Different letters indicate statistical differences ($p < 0.05$).

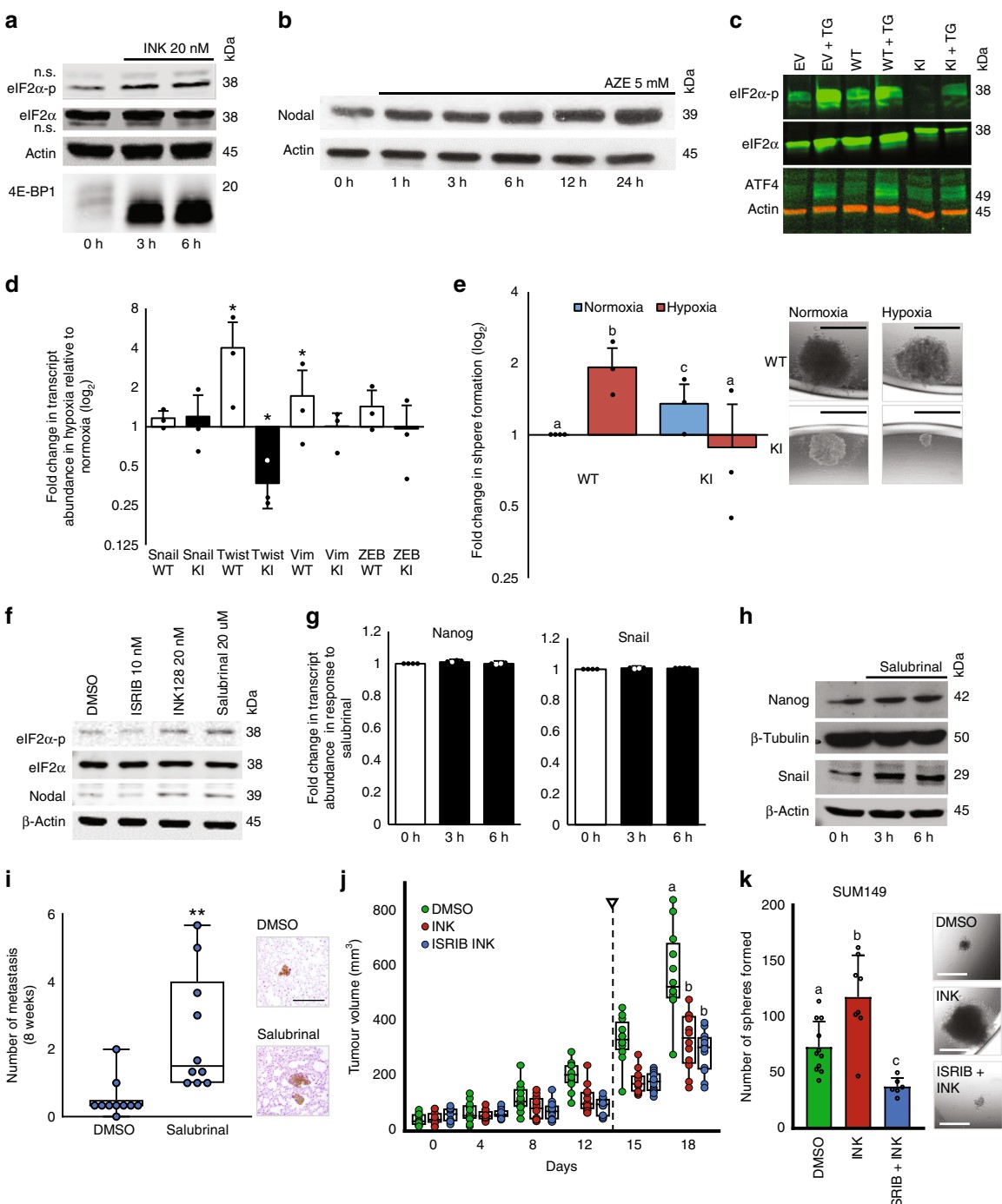

First, we activated the ISR with azetidine-2-carboxylate (AZE; 5 mM) for 1–24 h in T47D and MCF7 cells. AZE induced NODAL protein levels within 6 h of exposure by 2-to-4-fold (Fig. 5b; Supplementary Fig. 5c). Next, we depleted endogenous eIF2α in MDA-MB-231 cells and then expressed either a wild type (WT) or a non-phosphorylatable mutant (S51A)[48]. The expression of the S51A eIF2α mutant prevented up-regulation of ATF4 protein in response to TG (0.1 μM; 6 h) (Fig. 5c; Supplementary Fig. 5d). Exposure of MDA-MB-231 to 0.5% $O_2$ for 24 h increased the expression of stem cell-associated genes and sphere-forming frequency in cells expressing the WT but not mutant eIF2α (Fig. 5d, e). The spheres generated by S51A eIF2α cells were 70% smaller in normoxia and 90% smaller in hypoxia compared to WT cells (Supplementary Fig. 5e). This suggests that eIF2α phosphorylation

is needed for the hypoxic induction of breast cancer plasticity and stem-cell-like phenotypes.

To corroborate these findings, we treated T47D breast cancer cells for 24 h with salubrinal (which increases phospho-eIF2α by inhibiting PP1/GADD34) and then measured the expression of stem cell and EMT markers. Salubrinal (10 μM) induced NODAL protein levels in T47D cells (Fig. 5f; Supplementary Fig. 5f). Conversely, NODAL levels were reduced by ISRIB (10 nM), which allows efficient TC recycling even during ISR induction[42,43] (Fig. 5f; Supplementary Fig. 5f). Using qRT-PCR and immunoblotting we determined that compared to vehicle, salubrinal (up to 6 h treatment) induced NANOG and SNAIL protein, but not mRNA, in SUM149 cells (Fig. 5g, h; Supplementary Fig. 5g, h). As an in vivo extension, we exposed

**Fig. 5 Stress-induced plasticity is mediated by the ISR. a** eIF2α-p, eIF2α, and 4E-BP1 immunoblots from 0, 3, or 6 h INK (20 nM) treated T47D cells ($n = 3$). **b** NODAL immunoblot of 0–24 h AZE (5 mM) treated T47D cells ($n = 3$). **c** Control MDA-MB-231(EV) and MDA-MB-231 cells depleted for endogenous eIF2α and stably expressing wild type eIF2α (WT) or HA-tagged eIF2α with the S51A mutation (knock-in, KI) were treated with vehicle or TG. Immunoblots for eIF2α-p and ATF4 with eIF2α and β-Actin loading controls. **d** Mean $\log_2$ fold change of qRT-PCR quantification of *TWIST*, *ZEB1*, *SNAIL*, *VIM* transcripts in MDA-MB-231 WT (white) and KI (black) cultured for 24 h in normoxia or hypoxia ($n = 3$). **e** Mean sphere counts for MDA-MB-231 WT and KI pre-exposed to hypoxia or normoxia for 24 h relative to normoxia MDA-MB-231 WT spheres ($n = 3$) with representative images. **f** eIF2α-p, eIF2α and NODAL immunoblots from 24 h vehicle (DMSO), ISRIB (10 nM), INK128 (20 nM) or Salubrinal (20μM) treated T47D cells with β-actin and eIF2α loading controls ($n = 3$). **g** Mean $\log_2$-fold change of *NANOG* and *SNAIL* mRNA levels by qRT-PCR in Salubrinal-treated T47D cells (0, 3, or 6 h with (20 μM)) ($n = 4$). **h** NANOG and SNAIL immunoblot analyses of 0, 3, or 6 h Salubrinal (20 μM)-treated T47D cells with β-actin and β-tubulin loading controls ($n = 3$). **i** Mouse lung colonies 8 weeks following tail vein injection of SUM149 cells pretreated for 24 h with DMSO or Salubrinal (20 μM) ($n = 10$) with representative lung colony images (brown). **j** Mean SUM149 tumor volumes in mice treated every second day for 2 weeks (DMSO, INK (30 mg/kg) or INK + ISRIB (2.5 mg/kg)). Arrow indicates treatment cessation ($p < 0.01$, $n = 7$). **k** Mean sphere counts from 96,000 SUM149 cells dissociated from endpoint tumors from **j** ($n = 12$) with representative spheres images. Micron bars = 500 μm. Data represents independent experiments. Error bars indicate mean ± SD. Box and whisker plot represents median, interquartile range (IQR), whiskers extend to maximum and minimum. Two-sided *t*-test for paired samples. The asterisks denote *p*-values < *0.05, **0.01. Multiple comparisons tested by ANOVA. The same letters indicate relationships with a $p \geq 0.05$. Different letters indicate statistical differences ($p < 0.05$).

SUM149 cells to salubrinal for 24 h and then injected them into the lungs of mice through the tail vein. We quantified metastases 8 weeks later using HLA staining and determined that pre-exposure to salubrinal increased the initiation of metastatic colonies similar to that observed in response to INK (Figs. 5i and 4g), suggesting that mTOR and ISR crosstalk facilitates the emergence of breast cancer cell plasticity.

To assess mTOR and ISR crosstalk during the acquisition of BCSC phenotypes we treated mice harboring SUM149 xenograft tumors with vehicle (DMSO), INK128 (30 mg/kg) or INK128 (30 mg/kg) and ISRIB (2.5 mg/kg). Mice were dosed every second day for 2 weeks, starting when tumor diameter reached 5 mm. IHC analysis confirmed that phospho-4E-BP1 levels were inversely correlated with hypoxia (delineated by CA9) and that they were reduced in mice treated with INK128 as compared to the controls (Supplementary Fig. 5i). ATF4 levels were induced by INK128 (indicating ISR induction) and this was abrogated by ISRIB (Supplementary Fig. 5j). In the SUM149 xenograft model, tumor growth was reduced by both treatments (Fig. 5j). Cancer cells dissociated from tumors grown in INK128-treated animals had higher BCSC frequencies than those treated with vehicle, and ISRIB co-treatment abolished these increases (Fig. 5k). Since ISRIB reduced BCSC frequencies without reducing tumor growth, it is likely that it specifically targets the acquisition of BCSCs in this model. Collectively, these data suggest that the promotion of BCSC phenotypes by mTOR inhibition is partially mediated by the ISR and that ISRIB may improve therapeutic responses and disease outcomes by mitigating plasticity induced by hypoxia or TOR inhibitors.

**ISRIB abrogates chemotherapy-induced plasticity.** Chemotherapy has been shown to induce BCSCs[49]. We observed that paclitaxel (20 nM), a first-line breast cancer chemotherapy, increased eIF2α phosphorylation, while reducing phospho-4E-BP1 levels in T47D and SUM149 cells (Fig. 6a; Supplementary Fig. 6a). Moreover, paclitaxel-induced expression of stem cell and EMT-associated proteins NANOG, SNAIL, and SLUG at least two-fold within 3–6 h of treatment (Fig. 6b; Supplementary Fig. 6b). This confirmed that like hypoxia or INK128, paclitaxel induces the ISR and suppresses mTOR signaling, coincident with the acquisition of stem-cell-like phenotypes.

Considering its mitigating effect on hypoxia-induced or INK128-induced breast cancer plasticity, we assessed whether ISRIB improves the efficacy of paclitaxel. Testing varying concentrations of paclitaxel (2.5–50 nM) in the presence or absence of ISRIB (10 nM) on T47D and SUM149 cells, we determined that ISRIB sensitizes both cell lines to paclitaxel

(Fig. 6c; Supplementary Fig. 6c) as measured by colony formation assays. As an extension of our cell culture studies, mice bearing SUM149 mammary fat pad tumors were treated with vehicle (DMSO), ISRIB (2.5 mg/kg IP every second day), paclitaxel (15 mg/kg IV, weekly), or paclitaxel/ISRIB combination therapy for 2 weeks beginning when tumors reached 5 mm in diameter. IHC on tumor sections confirmed that paclitaxel increased ATF4 levels; this was reversed by ISRIB (Supplementary Fig. 6d). All treatment groups exhibited a reduction in tumor growth, with paclitaxel markedly reducing tumor burden with or without ISRIB (Fig. 6d). Cancer cells dissociated from tumors grown in paclitaxel-treated animals had higher frequencies of stem-cell-like cells than those treated with vehicle; an effect that was mitigated by ISRIB (Fig. 6e). Due to toxicities, breast cancer patients are often treated with suboptimal doses of paclitaxel. To determine whether ISRIB improves the efficacy of suboptimal chemotherapy, mice bearing SUM149 mammary fat pad tumors were treated with vehicle (DMSO), ISRIB (2.5 mg/kg IP), paclitaxel (20 mg/kg IP), or paclitaxel/ISRIB combination for 2 weeks beginning when tumors reached 5 mm in diameter. Using this paradigm, paclitaxel and ISRIB reduced tumor growth to the same extent, and ISRIB improved the efficacy of paclitaxel (Fig. 6f). Moreover, histological examination demonstrated that ISRIB reduced cell viability within the necrotic regions of these tumors (Fig. 6g). We confirmed these results using another triple-negative breast cancer model (MDA-MB-231). Mice bearing MDA-MB-231 tumors were treated with vehicle (DMSO), paclitaxel (15 mg/kg IV), or paclitaxel/ISRIB (2.5 mg/kg IP) combination therapy. Tumor volumes for drug-treated mice were calculated at 2 weeks. DMSO-treated mice reached endpoint (750 mm³) before therapy was completed. Notably, ISRIB improved the efficacy of paclitaxel in this model (Fig. 6h). This aligns with a recent study showing that ISRIB is particularly effective at reducing tumor burden in metastatic prostate cancers, characterized by high levels of phospho-eIF2α[50].

We utilized clinically relevant PDX models to determine whether ISRIB improves the efficacy of paclitaxel. Mice were treated with paclitaxel IV and ISRIB via gavage as recently described[50]. We established PDX401 (well-differentiated triple-negative) and PDX574 (poorly differentiated triple-negative) xenografts in the mammary fat pads of mice. We chose PDX574 because it contains high levels of hypoxia (marked by CA9) that co-localizes with NODAL, in contrast to the well-differentiated PDX401, which contains little hypoxia and lower levels of NODAL (Supplementary Fig. 6e) to account for both high and low BCSC-containing tumors. When tumors reached 5 mm in diameter, mice were treated with DMSO, ISRIB (10 mg/kg orally

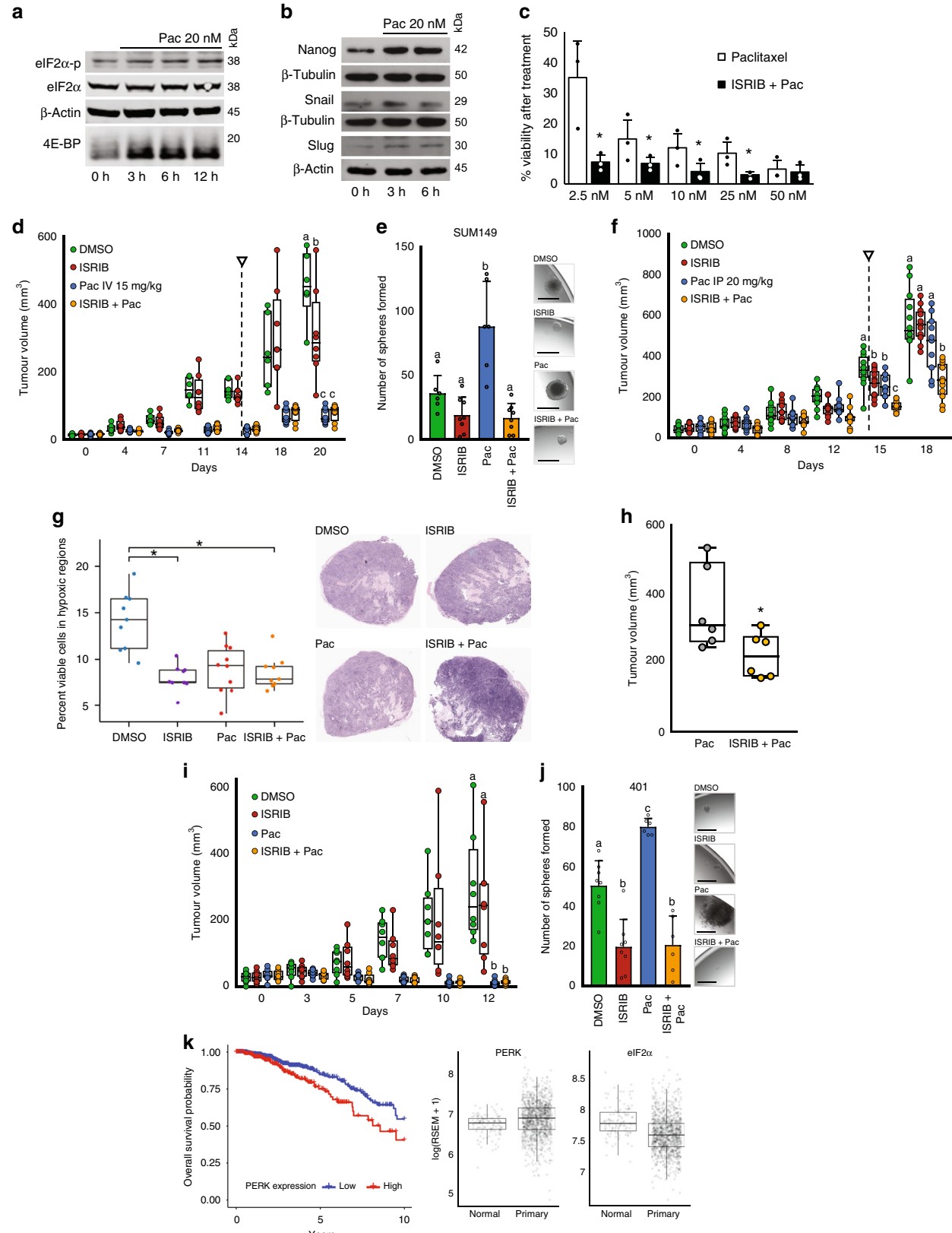

daily for 2 weeks), paclitaxel (15 mg/kg IV weekly for 2 weeks), or ISRIB and paclitaxel. IHC on endpoint tumors confirmed that ATF4 levels were increased by paclitaxel, whereas ISRIB prevented these effects (Supplementary Fig. 6f, g). ISRIB improved the efficacy of chemotherapy in paclitaxel-resistant PDX574 tumors (Supplementary Fig. 6h). Cancer cells dissociated

from PDX574 tumors grown in paclitaxel-treated animals had higher BCSC frequencies than those treated with vehicle, and this effect of paclitaxel was dramatically attenuated by ISRIB (Supplementary Fig. 6i). In turn, paclitaxel completely eradicated tumor growth in paclitaxel-sensitive PDX401-bearing mice (Fig. 6i). Nonetheless, cancer cells dissociated from PDX401

**Fig. 6 ISRIB mitigates therapy-induced BCSCs and improves efficacy of paclitaxel. a** eIF2α-p, eIF2α, and 4E-BP1 immunoblots from 0 to 12 h paclitaxel (20 nM)-treated T47D cells ($n = 3$). **b** Immunoblots of NANOG, SNAIL, and SLUG from 0, 3, or 6 h paclitaxel (20 nM) treated T47D cells ($n = 3$). **c** Mean percent viability of paclitaxel (2.5–50 nM)-treated T47D cells in the presence or absence of ISRIB (10 nM) ($p < 0.05$, $n = 3$). **d** Mean SUM149 tumor volumes in mice treated every second day for 2 weeks (DMSO, ISRIB (2.5 mg/kg IP), paclitaxel (15 mg/kg IV), or paclitaxel (15 mg/kg IV) + ISRIB (2.5 mg/kg IP)). Arrow indicates treatment cessation ($p < 0.01$, treatment $n = 8$, DMSO $n = 6$). **e** Mean sphere counts from 96,000 viable SUM149 cells dissociated from endpoint tumors in **d** ($n = 6$) with representative images. Micron bars = 250 μm. **f** Mean SUM149 tumor volumes in mice treated every second day for 2 weeks (DMSO, ISRIB (2.5 mg/kg IP), paclitaxel (20 mg/kg IP), or paclitaxel (20 mg/kg IP) + ISRIB (2.5 mg/kg IP)). Arrow indicates treatment cessation ($p < 0.01$, $n = 10$). **g** Quantification of viable cells within necrotic areas of H&E-stained SUM149 tumors from **f** ($p < 0.01$, $n = 7$) with representative images. **h** Mean MDA-MB-231 tumor volumes of mice given 2 weekly paclitaxel (15 mg/kg IV), or paclitaxel (15 mg/kg IV) + ISRIB (2.5 mg/kg IP) treatment. **i** Mean PDX401 tumor volumes in mice receiving the treatment from **d** ($n = 6$). **j** Mean sphere counts from 96,000 PDX401 cells dissociated from endpoint tumors from **i** ($n = 6$) with representative images. Micron bars = 250 μm. **k** Kaplan–Meier plot showing that low PERK expression correlates with survival (HR 1.80 (1.24–2.62), $p = 0.0019$). RNAseq expression of PERK (eIF2αK3) transcript in tumors relative to normal breast tissue (normal, $n = 113$; tumor, $n = 1062$; PERK $p = 1.52 \times 10^{-15}$, eIF2a $p = 1.87e-10$). Data represents independent experiments. Error bars indicate mean ± SD. Box and whisker plot represents median, interquartile range (IQR), whiskers extend to maximum and minimum. Two-sided t-test for paired samples. The asterisks denote *p-values < 0.05. Multiple comparisons tested by ANOVA. The same letters indicate relationships with a $p \geq 0.05$. Different letters indicate statistical differences ($p < 0.05$).

tumors grown in paclitaxel-treated animals exhibited higher BCSC frequencies than those treated with vehicle, again ISRIB attenuated this effect (Fig. 6j). Notably, these alterations in BCSC frequencies occurred independent of tumor growth differences, suggesting that ISRIB selectively prevents the acquisition of BCSC-associated phenotypes.

Collectively, these models demonstrate ISRIB's potential to attenuate the breast cancer plasticity induced by stimuli including hypoxia, mTOR inhibitors, and paclitaxel. Accordingly, analysis of RNA sequencing data from 1100 breast cancer patients in the TCGA demonstrates that high levels of PERK, which phosphorylates eIF2α during the ISR, are predictive of poor survival for 10 years following diagnosis with a hazard ratio of ~1.8 (Fig. 6k).

## Discussion

Stresses like hypoxia are an intrinsic aspect of the tumor microenvironment that are further amplified by chemotherapy. These stresses induce protective translational reprogramming, driving plasticity. In attempting to understand the mechanisms by which stem cell factors are selectively translated, we observed the existence of multiple mRNA isoforms (each with different 5′UTR features) derived from unique TSSs. The translation of a subset of the mRNA isoforms of NANOG, SNAIL, and NODAL promoted breast cancer cell plasticity in response to hypoxia, mTOR inhibition and chemotherapy by allowing efficient translation and escape from stress-induced translational repression (Fig. 7). The unique 5′UTR sequences of mRNA isoforms confer distinct sensitivities to alterations in translational machinery, which leads to differential translation of specific isoforms under stress conditions. Directly inhibiting pluripotency factors like NODAL, or interfering with the ISR, can circumvent this adaptation. These data establish a mechanism by which microenvironmental stresses induce tumorigenic phenotypes and the accumulation of cancer stem-cell-like properties enabled the dynamic selection of 5′UTRs. It will be interesting to determine the pervasiveness of this phenomenon and to establish whether transcriptional start site selection may alter the ratios of each transcript.

Several studies have shown that mTOR activation (associated with high protein synthesis rates) can also promote the expression of breast cancer plasticity factors involved in EMT[51]. We posit that subsequent to their initial induction, 5′UTR isoforms which are translated well in unstressed cells are transcriptionally sustained by positive feedback loops. For example, once induced by hypoxia, NODAL is able to sustain its own expression, even after reoxygenation[8]. Transitioning between highly proliferative (mTOR high; unstressed) and relatively dormant (mTOR low; stressed) states is likely requisite for tumor growth and plasticity,

respectively (Fig. 7). mTOR inhibitors have been utilized clinically[52]. Rapamycin and its analogs have been approved for treatment of cancer patients, including breast cancer patients and newer active-site mTOR inhibitors are undergoing clinical trials[44]. In this study, we reveal that mTOR inhibitors unexpectedly induce breast cancer cell plasticity by enabling the translation of NANOG, SNAIL, and NODAL isoforms in a manner similar to hypoxia. Much like chemotherapies, mTOR inhibitors would be expected to reduce tumor growth and metastasis; however, chronic treatment may result in the unintended propagation of a more aggressive disease, by fueling the accumulation of BCSCs. This study demonstrates that mitigating the ISR with a compound such as ISRIB can prevent the acquisition of BCSC phenotypes. Given the role of BCSC plasticity in therapy resistance and metastasis, the administration of ISRIB may improve outcomes in breast cancer patients treated with mTOR inhibitors or chemotherapy.

In conclusion, we highlight a mechanism of translational regulation whereby plasticity-inducing factors are synthesized during stress by expression of multiple mRNA isoforms harboring diverse 5′UTR sequences. We have further demonstrated that inhibiting this process (using ISRIB) may prevent the acquisition of BCSCs in hypoxic tumors or in response to therapy, so that adaptive therapy resistance and metastatic dissemination may be prevented.

## Methods

**Cell culture and treatments**. T47D, MCF7, and MDA-MB-231 cells, obtained from ATCC (Manassas, VI, USA), were maintained in RPMI-1640 Medium (Life Technologies; Carlsbad, CA, USA) with 10% fetal bovine serum (FBS) (Life Technologies). Cells were passaged using 0.25% (w/v) Trypsin (Life Technologies) as per ATCC recommendations. SUM149 cells, purchased from Bioreclamation IVT, were grown in Ham's F-12 medium with 5% heat-inactivated FBS, 10 mM HEPES, 1 μg/mL hydrocortisone, 5 μg/mL insulin. Breast cancer cells were authenticated at the Sick Kids Research Institute and tested for mycoplasma in house. H9 hESCs from WiCell (Madison, WI, USA) grown on irradiated CF-1 Mouse Embryonic Fibroblasts (GlobalStem; Gaithersburg, MD, USA) in knockout DMEM/F12 (Life Technologies; Carlsbad, CA, USA), 20% knockout serum replacement (Life Technologies), 1X non-essential amino acids (Life Technologies), 2 mM glutamine (Life Technologies), 0.1 mM 2-mercaptoethanol (BME; Thermo Fisher Scientific; Waltham, MA, USA), and 4 ng/mL of basic fibroblast growth factor (FGF) (Life Technologies). For experiments, cells were passaged into feeder-free conditions. Feeder-free conditions consisted of Geltrex matrix (Life Technologies) as a growth substrate and mTeSR1 media (Stem Cell Technologies; Vancouver, British Columbia, Canada). All cells were grown in a humidified environment at 37 °C with 5% $CO_2$.

**Hypoxia**. Hypoxia was administered at the noted concentrations using the Xvivo system (BioSpherix; Parish, New York, USA). Temperature (37 °C) and $CO_2$ (5%) were maintained. Time in culture was controlled for by placing cells in hypoxia at different times so that all samples were analyzed the same time after seeding. For

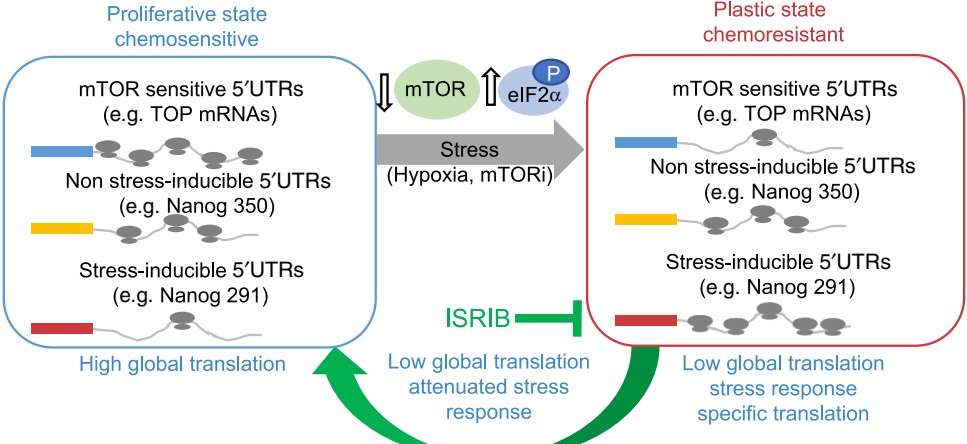

**Fig. 7 Differential 5′UTR utilization facilitates breast cancer plasticity.** A diagram of a working model demonstrating how the expression of multiple mRNAs sharing the same coding sequence but differing in their 5′UTR sequences can facilitate adaptation to insult—hypoxia, or mTORi. Though simplified, in response to stress mRNAs fall into three broad categories: Those whose translational efficiency is decreased in response to stress—mTOR sensitive 5′ UTRs; those with unaltered expression against a background of translational repression—non-stress-inducible 5′UTRs; and those translationally upregulated during stress—stress-inducible ATF4-like 5′UTRs. Alterations in the balance of expression resulting from this translational switch transition cells from a proliferative chemosensitive state to a plastic chemoresistant state. ISRIB blocks selective translation mediated by the induction of ISR, reducing the pro-survival plastic effects of the selectively translated mRNA.

example, 0 h cells were cultured for 24 h in normoxia, 6 h hypoxic cells were cultured for 18 h in normoxia followed by 6 h in hypoxia and 24 h hypoxic cells would be cultured for 24 h in hypoxia.

**Manipulation of NODAL.** To increase NODAL signaling, we used a NODAL expression vector (versus an empty pcDNA3.3 vector; pcDNA™3.3-TOPO® cloning kit; Invitrogen. Transfection was performed with Lipofectamine 2000 (Invitrogen) as per manufacturer instructions. For stable selection, Geneticin (G418; 800 ng/mL) was used[22,53]. We also employed recombinant human NODAL (rhNODAL; R&D). To decrease Nodal signaling, we used NODAL-targeted shRNAs (versus scrambled control shRNAs) Transfection was performed with Lipofectamine 2000 (Invitrogen) as per manufacturer instructions. For stable selection, Puromycin (200–450 ng/mL) was used[22,53]. To inhibit NODAL signaling, we also used SB431542. SB431542 selectively inhibits activin and TGF-β and NODAL signaling but not BMP signaling.

**Manipulation of mTOR and 4E-BP1.** To inhibit mTOR kinase activity we used MLN0128/INK128 (INK; 20 nM). To increase 4E-BP1 levels we used an expression vector (pCMV6-Entry EIF4EBP1 True ORF Gold Vector; OriGene). To knock down 4E-BP1 we used pGFP-V-RS EIF4EBP1 Human shRNA (OriGene) versus scramble and control vector. Transfection was performed with Lipofectamine 2000 (Invitrogen) as per manufacturer instructions.

**Manipulation of ISR components.** To induce ER stress, azetidine (AZE; 5 mM) or thapsigargin (TG, 0.1 μM) were used. Paclitaxel (Pac; 20 nM) was also used. In order to examine the role of eIF2α phosphorylation in the hypoxic induction of BCSC phenotypes, control (EV) MDA-MB-231 cells or cells wherein eIF2α was knocked down and then replaced with either a WT eIF2α or HA-tagged mutant eIF2α S51A (knock-in, KI).These cell were generated by infecting cells with infected with pMSCV retroviruses containing; sorting of GFP-positive cells by FACS; followed by renifection with pGIPZ lentiviruses expressing an shRNA specific for the 3′ UTR of human eIF2α mRNA; and selection in 2.5 μg/mL pur-omycin. In order to prevent eIF2α dephosphorylation, salubrinal (20 μM) was used. In order to overcome the effect of eIF2α on ternary complex turnover and trans-lation, the ISRIB (10 nM) was used.

**Immunoblotting.** Cells were lysed on-plate using Mammalian Protein Extraction Reagent (M-PER; Thermo Scientific), with Halt protease inhibitor cocktail (Thermo Scientific), and phosphatase inhibitor (Thermo Scientific). Protein was quantified according to manufacturer's instructions utilizing Pierce BCA Protein Assay Kit (Thermo Fisher) and measured on a FLUOstar Omega plate reader (BMG LABTECH; Offenburg, Germany). 4× Laemmli buffer (Bio-Rad; Hercules, CA, USA) with 5% BME (Sigma-Aldrich; St. Louis, MO, USA) containing 20 μg of protein was boiled for 10 min and loaded to be analyzed. Samples were separated by SDS–polyacrylamide gel electrophoresis, and then transferred onto Immobilon-FL membranes (Millipore). Precision Plus Protein Dual Color Standards (Bio-rad) was used to approximate molecular weight. Membranes were blocked with 5% milk in PBS 0.1% Tween (Sigma-Aldrich) for 1 h at room temperature, then incubated

with primary antibody overnight at 4 °C (Supplementary Table 2). After washing in PBS 0.1% Tween (Sigma-Aldrich), membranes were incubated with horseradish peroxidase-conjugated secondary antibodies (Bio-Rad) and then washed to remove excess secondary antibody. Clarity Western ECL Substrate (Bio-Rad) was used to detect signal. ChemiDoc™ XRS + System (Bio-Rad) or film were used to image the immunoblots. Densitometry was performed using ChemiDoc™ XRS + System (Bio-Rad).

**Florescence immunolot detection.** Using the Trans Blot Turbo (settings of 25 V and 1.3 A for 15 min; Bio-rad) proteins were transferred to a low-auto-fluorescence PVDF membrane (Bio-rad), blocked for 1 h at room temperature with Odyssey Blocking Buffer (Li-Cor; Lincoln, NE, USA), then incubated with primary antibody overnight at 4 °C in Odyssey Blocking Buffer with 0.1% Tween-20 (Sigma-Aldrich). Membranes were then probed with corresponding Li-Cor anti-mouse or anti-rabbit fluorescent secondary antibodies for one hour at room temperature at dilutions of 1/10,000 in Odyssey blocking buffer with 0.1% Tween-20 (Sigma-Aldrich) and 0.1% Tween. Imaging was conducted using the Li-Cor Odyssey Clx imaging system. Scans were performed at intensities that did not result in any saturated pixels.

**Densitometry.** Densitometry was performed using ImageJ. Each lane was isolated using the rectangle tool. Next, the background was established for each lane, and then the area of the resultant adjusted peaks was measured. Samples were nor-malized to control genes and measured against their respective untreated or vehicle controls as indicated in legends.

**Polysome profiling.** Cells were grown to 60–80% confluency at which time 0.1 mg/mL of cycloheximide was added to cells for 5 min at 37 °C before harvesting. The cells were extracted in polysome lysis buffer (15 mM Tris–HCl (pH 7.4)/15 mM MgCl$_2$/0.3 M NaCl/1% Triton X-100/0.1 mg/mL cycloheximide/100 units/mL RNasein), and the volume of each lysate to be loaded onto gradients was determined by total RNA. Sucrose gradients (7–47%) were centrifuged at 39,000 rpm (187,053×g) with a SW41-Ti Rotor (Beckman Coulter, Fullerton, CA) for 90 min at 4 °C. Gradients were continuously monitored at an absorbance of 254 nm and fractionated with a Brandel BR-188 Density Gradient Fractionation System. Each gradient was collected into nine equal fractions. The baseline absorbance of the sucrose gradient was calculated from the absorbance of a blank gradient using Peakchart software and subtracted from the absorbance reading of each sample. RNA isolation was conducted by first digesting each fraction with proteinase K, and extracting total RNA by phenol–chloroform extraction and ethanol precipitation. Samples were pooled into groups representing monosomes, light (<3 ribosomes) and heavy (more than three ribosomes) polysomes. Equal amounts of RNA were analyzed by real-time (RT) PCR. For validation studies, mRNA in polysomes (more than three ribosomes) was compared to total mRNA levels. For polysome profiles, the percentage of transcript in each fraction was calculated.

**RNA extraction and RT-PCR.** The PerfectPure RNA Cultured Cell Kit (5-Prime; Hilden, Germany) was used to extract total RNA from cultured cells following the

manufacturer's protocol. Optional DNase treatment was performed, and RNA was eluted in 50 µL. 3 µL of purified RNA was used for quantification using the Epoch plate reader (Biotek; Winooski, Vermont, USA). cDNA was made from purified total RNA using high capacity cDNA reverse transcription kit (Applied Biosystems; Foster City, CA, USA) as per manufacturer's protocol. The included random hexamers were used to prime reverse transcription and 1 µg of RNA was used for each. 'No Template' reactions did not contain RNA and 'No RT' reactions did not contain reverse transcriptase enzyme. RT PCR analysis was performed on 1 µL of cDNA using TaqMan Gene Expression Master Mix according to the manufacture's procedures using FAM labeled TaqMan® and PrimePCR gene expression human primer/probe sets (Thermo Scientific and BioRab; see Supplementary Table 3). mRNA expression was compared to untreated control using the ΔCT method. Data was collected on a CFX96 Touch Real-Time PCR Detection System (Bio-Rad; Hercules, CA, USA) using standard RT PCR settings. Activation 95 °C 10 min; Melting 95 °C 15 s; Annealing/extension 55 °C 1 min. Return to step 2 for 40 total cycles. Melt curve analysis was performed to ensure the production of a single amplicon. For absolute quantification of NODAL, after cDNA synthesis RT PCR was performed using Power SYBR Master Mix (Life Technologies). 1 µL of the cDNA was loaded in triplicate for quantification of NODAL (1 µL = 50 ng of starting RNA). The following primers were used: NODAL forward primer TACATCCAGAGTCTGCTG; and NODAL reverse primer CCTTACTGGATTA GATGGTT. Cloned NODAL PCR products were linearized, and diluted series was made (copy number/µL). A standard curve was constructed from these samples and run with the samples to estimate the number of NODAL transcripts at each time point. For isoform-specific PCR all products were validated using sequencing. The following primers were used: NANOG 350 forward primer GAT GGG GGA ATT CAG CTC AGG; NANOG 350 reverse primer TCA AGA CTA CTC CGT GCC CA; NANOG 291 forward primer AAC GTT CTG GAC TGA GC; NANOG 291 reverse primer AGG CAG CTT TAA GAC TTT TCT GG; SNAIL 417 forward primer AAA GGG GCG TGG CAG ATA AG; SNAIL 417 reverse primer CGC CAA CTC CCT TAA GTA CTC C; SNAIL 85 forward primer CGG CCT AGC GAG TGG TTC; SNAIL 85 reverse primer CAC TGG GGT CGC CGA TTC; NODAL small (42 + 14 + 298) forward primer CTG GAG GTG CTG CTT TCA GG; NODAL small (42 + 14 + 298) reverse primer CAG GCG TGC AGA AGG AAG G; NODAL 298 forward primer GTT TGG TAC CTA GAG CAG G; NODAL 298 reverse primer TCC AGG GAC GGG ATC TAG G; NODAL 416 forward primer CCC TCG GCA TTC TCT TCC TG; NODAL 416 reverse primer ATC CCT GCC CCA TCC TCT C.

Droplet digital PCR (ddPCR) was used to measure the number of each target transcript in purified RNA samples from total RNA and polysomal RNA. 9 µL of diluted purified RNA samples, 10 µL of 2x ddPCR Supermix for Probes (No dUTP), and 1 µL of 20× TaqMan Gene expression primer probe was used. 20 µL of each reaction mix were pipetted into the sample wells of DG8 Cartridge (cat #). 70 µL of Droplet Generation Oil for Probes or EvaGreen was added to the oil wells, then covered with DG8 Gaskets. Droplets were generated using QX200 Droplet Generator. 40 µL of generated droplets were transferred to 96-well PCR plate. All reactions using 2x ddPCR Supermix for Probes utilized thermal cycling conditions of 95 °C (10 min), 94 °C (30 s), 60 °C (1 min), 98 °C (10 min), and 4 °C (infinite). Steps 2 and 3 were repeated 40 times. All ddPCR thermal cycling conditions used adjusted slow ramp rate of 2 °C/s. PCR plates were placed into QX200 Droplet Reader for the reading of droplet number and amplitude of fluorescence intensity.

**Sphere formation.** Sphere formation media was composed of DMEM/F12 + GlutaMax (Life Technologies), 1x B27 (Life Technologies), 20 ng/mL epidermal growth factor (EGF) (Life Technologies), and 10 ng/mL FGF (Life Technologies). After treatment, cells were harvested using 0.25% (w/v) trypsin (Life Technologies), the trypsin was neutralized, and the cells resuspended in fresh media. These cells were filtered through a 40 µm pore filter (Thermo Fisher) to obtain a single cell solution. Cells were counted using trypan blue and diluted in the sphere formation media to the appropriate concentration for plating. Cell concentrations were titrated to ensure fewer than 1 sphere per well for each set of experimental conditions. An equal number of cells were plated for each condition wherein 200 µL of the diluted cells were seeded into each well of a 96-well ultra-low attachment surface plate (Corning, NY, USA). Spheres were given between 10 and 21 days to grow. Sphere exceeding diameter >50 µm were count on an inverted microscope and the total number of spheres in each plate are reported. Images of spheres were captured using the EVOS FL Cell Imaging System (Thermo Fisher) at ×4 magnification. In order to enrich for spheres, cells were cultured in a bioreactor (Synthecon). One million cells in 10 mL RPMI + 10% FBS were loaded into a bioreactor (Synthecon Rotary Cell Culture System). Cell were grown in standard growth conditions rotating at ~7 rpm for 3 days[54]. Where sphere size was measured, ImageJ was used to threshold the spheres against the background and the total area of the sphere was acquired.

**Flow cytometry.** One million cells were stained in 100 µL of Zombie Aqua (Fixable Viability Kit BioLegend; San Diego, CA, USA) for 20 min at room temperature. Zombie aqua was removed and 20 µL of antibody dilution was added to each sample, which was then incubated on ice for 10–15 min.
Antibody pairs:

CD24 APC (REA832, MiltenyiBiotec, 1:20 dilution), CD44 Vioblue (REA690, Miltenyi Biotec, 1:5 dilution) used in Fig. 1f
FITC Mouse Anti-Human CD24 (BD Biosciences; Franklin Lakes, NJ, USA, 1:5 dilution), PE Mouse Anti-Human CD44 (BD Biosciences, 1:5 dilution) used in Fig. 3d and Supplementary Figs. 1b–d and 3f,g

Cells were washed with 200 µL FACs buffer (PBS with 1% FBS), pelleted and then resuspended in 100 µL 2% PFA in FACs buffer. For acquisition, cells were re-suspended in 300 µL FACS buffer for flow acquisition. Doublet discrimination and live cell gates were used to identify the cells of interest, and quadrant gates were set according to the fluorescence minus one controls (FMO) (Supplementary Fig. 7).

**RNA-seq and gene set enrichment.** RNA was extracted from hypoxia-treated cells with the Qiagen RNeasy kit and quantified via Nanodrop; quality was measured using Qubit. RNA was shipped to McGill University and Genome Quebec Innovation Center, where quality was validated via Bioanalyser, followed by NEB/KAPA library preparation and sequencing via Illumina HiSeq. Post sequencing quality check of reads was performed with FastQC and adapter sequences removed using Skewer. Reads were aligned to the GRCh37/Hg19 human reference genome using STAR. Data processing included Bigwig, PCA, correlation matrices, and coverage maps of aligned reads were produced using DeepTools. Read quantification via FeatureCounts was performed using Refseq annotations. Expression values for paired samples, hypoxia/normoxia, were obtained using the exact test within the edgeR package. An adjusted p-value (FDR) of 0.05 was used to determine statistically significant differences (p-value adjusted for multiple hypothesis testing by the Benjamini–Hochberg method). GAGE package for R was used to compare data to the hallmark gene sets from Molecular Signatures Database, which was used for Gene set analysis.

**Luciferase reporter assays.** For cloning of the luciferase 5′ UTR reporters, we modified the PTK-ATF4-Luc plasmid (containing the mouse ATF4 5′UTR) from Dey et al.[39] using the QuikChange Lightning Site-Directed Mutagenesis Kit (Agilent; Santa Clara, CA, USA) to introduce two BsmBI restriction sites flanking the ATF4 5′ UTR. The ATF4 5′ UTR was then replaced with a LacZ insert to complete the pGL3-TK-5UTR-BsmBI-Luciferase plasmid used for one-step cloning of various 5′ UTRs. This plasmid has been made available on Addgene (plasmid #114670). Except for "Snail 417", 5′ UTRs of interest flanked by adapter sequences for cloning were ordered as gBlocks from Integrated DNA Technologies (Coralville, IA, USA; sequences are listed in Supplementary Table 4) and cloned into pGL3-TK-5UTR-BsmBI-Luciferase using BsmBI (New England BioLabs; Whitby, ON, Canada). The "Snail 417" UTR insert was generated by PCR using the forward primer TATCGTCTCAACACCGAGCGACCCTGCATAAGCTTGGCGCTGAGC CGGTGGGCG and the reverse primer ATACGTCTCTCTTCCATAGTGGTC GAGGCACTGGGGTCG. The "NODAL 298" uORF mutant was generated using the QuikChange Lightning Site-Directed Mutagenesis Kit with the forward primer CCTCCGGAGGGGGGTTATATAATCTTAAAGCTTCCCCAG and the reverse primer CTGGGGAAGCTTTAAGATTATATAACCCCCCTCCGGAGG, to introduce a G-to-A mutation within an upstream start codon (ATG) at position -104 relative to the translational start. HEK293 cells were transfected using Polyethylenimine (PEI, Sigma Aldrich; St. Louis, MO, USA). PEI and vector media were combined in a ratio of 5 µL PEI and 0.5 µg vector in 250 µL of serum-free DMEM and incubated for 10 min at room temperature. 250 µL of the mixture was apportioned into each well of a 12-well plate containing 250 µL DMEM. HEK293 cells were incubated overnight in the transfection mixture. The transfection media was removed and replaced with DMEM 10% serum and cells were given 24 h to recover. Cells were then treated with 0.1 µM of thapsigargin (TG) for 3 h or 0.5% $O_2$ for 6 h (hypoxia). Upon completion of treatment, cells were lysed and the luciferase activity measured with the Firefly Luciferase Assay System (Promega, Madison, WI, USA). Activity was read using a FLUOstar Omega plate reader (BMG LABTECH).

**Animal models.** All experiments involving animals were approved by the Animal Use Subcommittee at the University of Alberta (AUP00001288 and AUP00001685). To avoid potential confounding variables associated with estrogen supplementation, and given that BCSC phenotypes appear to be commonly regulated irrespective of subtype, we chose to limit our in vivo studies to ER-negative models.

**Experimental metastasis assay.** SUM 149 cells were pre-incubated as described, then trypsinized and counted. 500,000 cells in 100 µL $Ca^{2+}$-free HBSS were injected into the tail vein of female NOD-scid IL2Rgamma$^{null}$ (NSG) mice. Mice were sacrificed at 8 weeks (to tumor formation). Lungs were formalin-fixed and paraffin-embedded, and IHC staining on this tissue was conducted using a human-specific HLA antibody (Supplementary Table 1) as per the manufacturer's instructions. For each mouse organ, 3–6 sections were acquired from evenly spaced areas throughout the tissue, and the average number of metastases per mouse organ was calculated.

**Orthotopic xenografts**. 500,000 SUM 149 or MDA-MB-231 cells in 100 μL RPMI: Matrigel (1:1) were injected into the right thoracic mammary fat pad of 7–8-week-old female NSG mice. Mice were randomized and treatments were administered when tumors reached a maximum diameter of 5 mm. At this point, mice were treated with DMSO vehicle control, INK (30 mg/kg by gavage), ISRIB (2.5 mg/kg IP), or paclitaxel (20 mg/kg IP or 15 mg/kg IV) for the times indicated. Tumor measurements were taken twice per week and a digital caliper was used to measure length × width × depth of the tumor upon excision in order to calculate volume. Mice were sacrificed when tumors reached ~1 cm in diameter. Tumors were cut in half. One half was dissociated and the other was fixed with 4% formaldehyde, paraffin embedded, sectioned, and stained with H&E or used for immunohistochemistry. Survival curves for overall survival were constructed using the Kaplan–Meier method and significance determined by log-rank test.

**Patient-derived xenografts**. Two PDX models obtained through a collaboration with Oncotest (Charles River, Freiburg, Germany) were used: PDX401 is a well-differentiated basal-like TNBC and PDX574 is a poorly differentiated basal-like TNBC. Viable pieces (~1 mm in diameter) were placed, through a small incision, into the mammary fat pads of 7–8-week-old female NSG mice. At this point, mice were treated with DMSO vehicle control, INK (30 mg/kg by gavage), ISRIB (2.5 mg/kg IP or 10 mg/kg by gavage) or paclitaxel (20 mg/kg IP or 15 mg/kg IV) for the times indicated. Tumor measurements were taken twice per week and a digital caliper was used to measure length × width ×depth of the tumor upon excision in order to calculate volume. Mice were sacrificed when tumors reached ~1 cm in diameter. One half was dissociated and the other was fixed with 4% formaldehyde, paraffin embedded, sectioned and stained with H&E or used for immunohistochemistry. Survival curves for overall survival were constructed using the Kaplan–Meier method and significance determined by log-rank test.

**Tumor dissociation**. Half of each tumor was dissociated using the Human Tumor Dissociation Kit and Gentle MACS Tissue Dissociator with Heaters (Miltenyi Biotec; Bergisch Gladbach, Germany) according to the manufacturer's instructions prior to enumeration of live cells using trypan blue.

**Analysis of tumor necrosis**. Three tumor sections spaced evenly throughout each tissue block were stained with H&E. Each tumor section was imaged such that the entire section was visible in one field of view. All IHC images were analyzed using ImageJ. Total tumor section area was selected and measured by thresholding against background pixel intensity. Necrotic regions are comprised of pixels of higher uniform intensity compared to the intact tumor cells and a second round of thresholding against the lighter necrotic regions was used to distinguish and measure the total area of necrosis. Viable cells in hypoxic regions were determined by identifying dense, dark areas within necrotic regions. These dark foci represent cells with intact nuclei and the area occupied by these cells was divided by the area of the necrotic region to capture percent viable cells in hypoxia regions.

**Immunohistochemistry**. Formalin-fixed, paraffin-embedded tissue underwent deparaffinization in xylenes, hydration through an ethanol series, antigen retrieval with citrate buffer, and peroxidase and serum-free protein blocking. NODAL, CA9, p4E-BP1, 4E-BP1, or ATF4 specific antibodies (Supplementary Table 2) were applied. Slides were rinsed in TBS-T, and treated with Envison+ HRP anti-mouse IgG (Dako). Color was produced with DAB (brown) substrate and counterstained with Mayer's haematoxylin. Samples were dehydrated in reagent grade alcohol and cover slipped with permanent mounting medium. Negative control reactions were conducted with mouse IgG, isotype controls used at the same concentration as the primary antibodies.

**Analysis of patient data**. Level 3 TCGA RNAseqV2 BRCA gene expression data and clinical information was obtained from the TCGA Data Portal in August 2014. RNA-sequencing RSEM values were used in downstream analyses. For TCGA RNA-seq samples, relative abundance (transcripts per million, TPM) was calculated by multiplying the scaled estimate data by 106 and used in downstream analysis.

We conducted all analyses and visualizations in the RStudio programming environment (v0.98.501). R/Bioconductor packages ggplot2, plyr, pROC, survival, GAGE, and limma were used where appropriate. 4E-BP1 and PERK expression was dichotimized with receiver operating characteristics (ROC) curves to determine the optimal cutoff for the endpoint of overall survival censorship. Quantitative differences between high versus low expression cohorts were evaluated with a Student's t-test; qualitative differences were evaluated using a Fisher exact test. Survival curves for overall survival were constructed using the Kaplan–Meier method and significance determined by log-rank test/Wilcoxon test.

**Surviving fraction assays**. Cells were incubated with paclitaxel ± ISRIB for 1 h in a standard $CO_2$ incubator and then washed with phosphate buffered saline (PBS) and made into a single cell suspension and plated. After 7–14 days, colonies were fixed with acetic acid–methanol (1:4) and stained with crystal violet (1:30).

**Statistics**. Analysis was conducted using GraphPad Prism 7. Student's t-test was employed for direct analysis of a single condition to the appropriate control with paired two-sample t-tests used where appropriate. p values < 0.05 were considered statistically significant (signified by *). To analyze the relationship between multiple conditions, one-way ANOVAs for all pairwise comparisons with the Bonferroni and Holm post-hoc test were employed to detect statistically significant differences between groups and correct for family-wise error rate (signified by letters). p-values < 0.05 were considered statistically significant.

**Reporting summary**. Further information on research design is available in the Nature Research Reporting Summary linked to this article.

## Data availability
The source data underlying Figs. 1a–n, 2d–k, 3b–f, 4a–k, 5a–k and 6a–j, and Supplementary Figs. 1a–l, 2b–d, 3a–d, 4a–l, 5a–j and 6a–d, f, h are provided as a Source Data file. RNA-sequencing data that support the findings of this study have been deposited in GEO with the accession code GSE149132 and will be made available upon publication of the article. All the other data supporting the findings of this study are available within the article and its supplementary information files and from the corresponding author upon reasonable request. A reporting summary for this article is available as a Supplementary Information file.

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

## Acknowledgements

This work was supported by the Canadian Institutes for Health Research (PLS 9538 and PLS 95381), the Canadian Breast Cancer Foundation Prairies, the Alberta Cancer Foundation, the Women and Children's Health Research Institute and Alberta Innovates Health Solutions through grants awarded to L.-M.P. S.D.F. has been supported by a doctoral scholarship from the Natural Sciences and Engineering Research Council of Canada, and an Ontario Graduate Scholarship. K.M.V. was a Vanier scholar, C.P. was a recipient of a studentship from the McGill Integrated Cancer Training Program and L.L. and M.C. were supported by Ph.D. scholarships from the AIHS and CIHR. L.-M.P. was the recipient of the premier new investigator award from the CIHR, the Sawin-Baldwin Chair in Ovarian Cancer, the Dr. Anthony Noujaim Legacy Oncology Chair and the AIHS translational health chair in cancer. I.T. is a scholar of the Fonds de Recherche du Québec-Santé (FRQS; Junior 2). This work was also supported by R-37-DK060596 and R01-DK 053307 awarded to M.H., and by CIHR MOP 38160 a Canadian Cancer Society Research Institute grant (CCSRI no. 700886) and a grant from the Quebec Breast Cancer Foundation awarded to A.K.

## Author contributions

L.-M.P. and M.J. conceived the project, wrote the manuscript and produced the figures. M.J. conducted immunoblots, RT-PCR assays, reporter assays, sphere formation assays, flow cytometry, and transfections. L.L. performed quantification of mRNA on polysomes, RNAseq analysis quantification of necrosis. G.Z. and J.L. performed all animal experiments and IHC. S.D.F. designed isoform-specific PCR assays and derived constructs for reporter assays; K.M.V. conducted analyses of patient data. K.T. performed and validated polysome profiles, D.D.-C. measured NODAL in cells grown in 3D. D.Q. conducted sphere formation assays related to NODAL. I.D. assisted with flow cytometry. M.C. and Z.X. assisted with immunoblotting and RT-PCR of stem cell genes. B.-J.G., M.H., M.L., A.P., A.B., and J.U. assisted with polysome fractionation and C.P. and A.K. provided the eIF2α KI expressing cells. J.S. supplied PDX models, G.M.S. assisted with editing, flow cytometry, and characterization of PDX models. I.T. assisted with project design and editing.

## Competing interests

The authors declare no competing interests.
