## [Peer Review File · Nature Communications]

Reviewers' comments:

Reviewer #1 (Remarks to the Author):

In this manuscript, the authors connect translational control with new strategies to overcome drug resistance to ultimately reduce metastatic breast cancer. They demonstrate that NODAL, NANOG, and SNAIL are differentially translated in hypoxic conditions which enrich for cancer stem cells in breast cancer cells. Most importantly, the authors need to establish a link between the importance of the new isoforms to the metastatic phenotype. At this stage, it is not clear the specific function of the isoforms because they are translated into a functional protein. Since there are not multiple NANOG, SNAIL or NODAL protein isoforms the relevance of the new transcripts induced by hypoxia is not clear. I believe this first part of the manuscript is weak and the authors failed to connect the importance of the translational reprogramming, as they called it, to the metastatic phenotypes, mTOR signaling and ISR. Hence, the manuscript reads as if two stories were glued together.

1. The authors need to show that blockade of specific NANOG, SNAIL or NODAL isoforms is biologically relevant, otherwise upregulation of NANOG, SNAIL or NODAL might just be mediated by gene regulation mechanisms. For instance, how is the expression of the NANOG, SNAIL or NODAL regulated during hypoxia? One plausible possibility is that hypoxia-specific transcription factors or epigenetic machineries are recruited to their promoter or cis-regulatory sites to induce their activation. If this is the case, expression of the different isoforms per se is completely irrelevant as the important end point is the accumulation of the protein.
2. The authors demonstrate in Figures 1-2 a correlation between upregulation of specific 5'UTR isoforms of NODAL, NANOG, and SNAIL and downregulation of mTOR during hypoxia, leading to an increase in CSC/EMT signatures. Figure 3 delves into functional outcomes of inhibiting mTOR in breast cancer cell lines in vitro and in vivo, however, the authors do not query the expression of NODAL, NANOG, or SNAIL nor their isoforms in these later assays. Why do the authors solely focus on the effects of different inhibitors/chemotherapy agents in the later figures without correlating them to the effects on the translation of these three factors? There seems to be a critical logical gap in the transition between Figures 1-2 and the remainder of the figures, and a discrepancy in the data presented with the manuscript title.
3. It is not clear why SNAIL transcripts are downregulated in SUM149 but upregulated in T47D after cells being in hypoxia for 6h, yet in both cells more SNAIL protein is accumulated.
4. What is the effect of depleting NODAL by shRNA in cells under hypoxia?
5. The authors need to discuss why the PDX and the SUM149-derived xenografts show opposite results in the INK condition (Fig. 3h and 4j).
6. Fig. 1h is not referred in the text. Fig. 3d: bar graph and FACS profiles do not match.
7. Treatment of ER+ breast cancer and TNBC patients is radically different. Along the manuscript experiments are performed using ER+ and TNBC cells without a proper justification and rationale. The authors should clearly indicate why some experiments were done with two different breast cancer subtypes and others only with one subtype.
8. Authors should be more specific when describing results, i.e., quantitative numbers should be included in parentheses where applicable.
9. Representative pictures of spheres throughout the manuscript are of one sphere to highlight difference in sphere sizes between treatment conditions, however, sphere-forming ability was demonstrated using quantifications of sphere number. Were sphere sizes also quantified? How was sphere counting performed? Specifically, was there a cutoff sphere size that was considered for

counting? The methods are not clearly detailed in the text nor the methods section.

10. Figures 3 and S3 is dedicated to demonstrating that mTOR inhibition increases the CSC/EMT signature of breast cancer cells.

a. In line with comment #7, all in vitro work was performed in non-highly metastatic cell lines T47D and MCF7. However, in vivo xenografts use the highly metastatic SUM149 line. The authors could elaborate, again, on the choice of cell lines for each assay.

b. Methods state that 700ul total volume (containing 5×10^5 cells) were injected into the tail veins of mice – more than double the maximum volume according to NIH guidelines. Is this a typo error?

c. The focus of this figure is to demonstrate the increase in CSC/EMT signatures upon treatment with INK, i.e., mTOR inhibition. INK inhibition in a PDX model also increases primary tumor growth (Figure 3h). The authors do not perform additional assays characterizing the primary tumors (only a sphere assay, Figure 3i), nor do they discuss in better detail how this result relates to their prior observations. Additionally, limiting dilutions xenografts would be better suited to demonstrate CSC frequencies in vivo after INK inhibition.

Reviewer #2 (Remarks to the Author):

In this manuscript the authors describe the selective translation of isoforms of NANOG, SNAIL and NODAL in breast cancer cells exposed to hypoxia. These findings suggest a model of translational regulation whereby differences in 5'UTR features of the mRNA isoforms harboring the same ORF allow isoform-specific translation and consequent upregulation of NANOG, SNAIL and NODAL proteins under hypoxia. This is considered by the authors as the induction of an adaptive stem cell program which leads to the acquisition of breast cancer cells with stem cell-like features (BCSC) phenotypes. This process was also induced by mTOR inhibitors and chemotherapeutics and was inhibited by an Integrated Stress Response Inhibitor (ISRIB).

Although this is an interesting manuscript several points remain stratification.

Fig. 1h: control SB normoxia is missing.

Fig. 1i: To better appreciate the exact role of hypoxia, adjusted normoxic controls, (f.e. 12h, 24h) are needed.

Fig 1k: Statistical analyses are missing. How many independent replicates were performed?

Fig 1n: Only one loading control (actin) is displayed. This most probably does not account for all WBs displayed. Please show appropriate loading controls for all experiments as well as quantification.

Fig. 3a: Quantification is missing.

Fig. 3b,c,d: SB alone experiments are missing.

Fig. 3f: Only one loading control (actin) is displayed. This most probably does not account for all WBs displayed. Please show appropriate loading controls for all experiments as well as quantification.

Fig. 4a: Statistical analyses are missing. How many independent replicates were performed?

Fig. 4b: To better appreciate the exact role of AZE, adjusted controls, (f.e. 12h, 24h) are needed.

Fig. 4e: If sphere size has been measured this does not seem to be increased by hypoxia. please specify approach.

Fig. 4g, h: Only one loading control (actin) is displayed. This most probably does not account for all WBs displayed. Please show appropriate loading controls for all experiments as well as quantification. The quality of Nanog and Sox2 WB is poor. Solvent controls are missing. Fig. 4h:

Correct typo with Salubrinal.

Fig. 5a,b: Only one loading control (actin) is displayed. This most probably does not account for all WBs displayed. Please show appropriate loading controls for all experiments as well as quantification. The quality of Sox2 WB is poor. Solvent controls are missing.

Fig. 5g: How exactly were viable cells determined? Histology suggests a difference between ISRIB and Pac which is not displayed in the graph.

Suppl Fig. 1j: No loading controls are displayed. No controls for 12h, 24h values are shown. This is also true for many subsequent figures in the supplement.

Overall, the number of independent replicates is not clear throughout the study which raises concerns regarding reproducibility and validity. In addition, although the use of patient derived xenografts is of interest, conclusions are drawn from one well-differentiated versus one poorly differentiated patient sample. More set-ups from different patients are needed to support the conclusions.

Furthermore it would be of interest to see whether the different gene isoforms are indeed present in different breast cancer patient populations and which of them relates to chemoresistance.

A scheme summarizing the main findings would be beneficial since the manuscript is at certain points lengthy and not very focused.

Reviewer #3 (Remarks to the Author):

This very interesting manuscript by Jewer et al. describes a mechanism of translational plasticity underlying breast tumor progression and its therapeutic responses. Specifically, in the first part of the manuscript, the authors report that breast cancer cells translate specific mRNA isoforms of NANOG, SNAIL and NODAL transcripts under hypoxic conditions. The authors further showed that mTOR inhibition as well as chemotherapeutic agents induce translation activation of NANOG, SNAIL and NODAL mRNAs, which in turn cause breast cancer cells to acquire stem cell-like phenotypes, ultimately contributing to tumor growth. Interestingly, this translation reprogramming downstream these stress agents are functionally linked to eIF2 α phosphorylation (P-eIF2 α). Importantly, ISRIB, which inhibits the activity of P-eIF2 α , represses tumor growth or the acquisition of stem-cell phenotypes in xenografts or PDX models treated with mTOR inhibitor treatments or paclitaxel.

This is an important paper that highlights the interplay between distinct pathways that converge on manipulating the translational machinery as a mechanism to drive and maintain tumor growth even in the presence of anti-cancer drugs. The manuscript is timely and deserves to be published; however, I have a few comments that should be addressed before publication.

-To tie the first part of the paper with the second, the authors should analyze the RNA level as well as the translation efficiency of the specific isoforms (NANOG, SNAIL and NODAL mRNAs) after mTOR inhibition, chemotherapeutics, and with or without ISRIB as they performed in Figure 2 upon hypoxia.

- There are several reports in literature showing that mTOR activation actually results in increased transcription and translation of EMT signatures, including SNAIL, Vimentin and NANOG. Therefore, the authors should clarify (in the discussion section) that the effect they observe on the translation upregulation of specific mRNAs with mTOR inhibition is very specific to these breast cancer cell types and treatment conditions. This is important as while the authors interestingly found that these mRNAs possess different isoforms with a distinct 5'UTR, it is unclear which of these isoforms

will eventually contribute to protein abundance in the cells. For example, can the authors mutate the regulatory uORF in the Nodal mRNAs and assess the protein level upon normoxia and hypoxia?

-In Figure 4, panel C should be quantified as it is not clear what the ATF levels are under different conditions.

- It would be interesting if the authors can assess (if it is possible) whether the increase of P-PERK and P-eIF2 α can predict for poor survival in breast cancer patients.

RESPONSE TO REVIEWS

We would like to thank the reviewers for the insightful comments which helped us to critically improve our manuscript. Please find below our point-by-point response to each of the comments. You will see that we have fully addressed these comments with experimental and/or editorial revisions. Our point-to-point answers to reviewers' comments and concerns are outlined below:

Reviewers' comments:

Reviewer #1 (Remarks to the Author):

“In this manuscript, the authors connect translational control with new strategies to overcome drug resistance to ultimately reduce metastatic breast cancer. They demonstrate that NODAL, NANOG, and SNAIL are differentially translated in hypoxic conditions which enrich for cancer stem cells in breast cancer cells. Most importantly, the authors need to establish a link between the importance of the new isoforms to the metastatic phenotype.”

We apologize to the reviewer for the lack of clarity. Several studies have shown that NODAL, NANOG and SNAIL contribute to metastatic phenotypes, and that stresses such as hypoxia similarly promote metastatic potential. Our findings help to explain how the aforementioned stem cell associated transcripts are translated during hypoxia, even when global protein synthesis is greatly diminished. The clinical significance has been established in the extant literature, and relates to NODAL, NANOG and SNAIL in general¹⁻³. Our results help establish a mechanism, and thus potential targetable vulnerability, for stress-induced plasticity, which has never been described before. This discovery stemmed from our initial observation (outlined in Figure 1) which was a discordance between mRNA levels and corresponding proteins which induce plasticity under hypoxia. We subsequently showed that this was due to the described phenomenon of selective translation (confirmed to occur in Figure 1). When trying to understand which 5'UTR features may be enabling selective translation, we unexpectedly determined that NODAL, NANOG and SNAIL mRNAs exist in multiple isoforms which differ in their 5'UTRs but contain the same ORFs and thus encode the same proteoforms. The presence of these different isoforms allows for translation under different conditions. For instance, isoform 298, but not 42 of NODAL mRNA allows translation to occur in hypoxia. This enables the maintenance or increase of NODAL protein levels during the initial exposure to hypoxia. We provide evidence for such unexpected regulation of NODAL and other plasticity factors in Figure 2. In the revised version of the manuscript, we clarified this to the best of our ability where we specifically refer to “mRNA isoforms”, and as outlined below provide further evidence to corroborate the aforementioned model.

“At this stage, it is not clear the specific function of the isoforms because they are translated into a functional protein. Since there are not multiple NANOG, SNAIL or NODAL protein isoforms the relevance of the new transcripts induced by hypoxia is not clear. I believe this first part of the manuscript is weak and the authors failed to connect the importance of

the translational reprogramming, as they called it, to the metastatic phenotypes, mTOR signaling and ISR. Hence, the manuscript reads as if two stories were glued together.”

Again, we apologize for the lack of clarity. We show that NANOG, SNAIL and NODAL accumulate in hypoxia because a subset of their mRNA isoforms which differ in their 5'UTR regions can still be translated (akin to e.g., ATF4 mRNA, used as a control), whilst the majority of transcripts cannot be translated (akin to e.g., β ACTIN mRNA, used as a control). This leads to sustained and/or increased expression of NANOG, SNAIL and NODAL proteins even when the total levels of mRNA are unchanged and/or suppressed. Using reporter assays and polysome profiling we show that this is most likely caused by the 5'UTR features present in some (but not all) NANOG, SNAIL and NODAL mRNA isoforms, which allows selective isoform translation under conditions when mTOR is inhibited and ternary complex is limiting (do to the integrated stress response (ISR)). In other words, relevance of these isoforms is that they can selectively be translated when exposed to stresses such as hypoxia, mTOR inhibitors and/or chemotherapies. This translational response explains how NANOG, SNAIL and NODAL protein levels can be maintained or increased when the total mRNA levels (sum of all mRNA isoforms; please see the answer to comment 1 below) are either reduced or unchanged, respectively. The first part of the manuscript shows that stem cell reprogramming factors are selectively translated in stress because they have a diversity of 5' leaders. The second half of the article shows new data that similar effects to hypoxia were observed upon mTOR inhibition. Functionally, we demonstrated that these effects on translation induce breast cancer stem cell phenotypes, and that this can be abrogated by ISRIB. Considering that NANOG, SNAIL and NODAL play a major role in cancer cell plasticity and metastatic spread, we trust that this unique and hitherto unappreciated mechanism of regulation of their levels is directly pertinent to their role in metastasis, and thus feel like the two parts of the manuscript are connected in a logically sound manner. We clarified this in the revised version of the manuscript with the inclusion of schematics.

“1. The authors need to show that blockade of specific NANOG, SNAIL or NODAL isoforms is biologically relevant, otherwise upregulation of NANOG, SNAIL or NODAL might just be mediated by gene regulation mechanisms. For instance, how is the expression of the NANOG, SNAIL or NODAL regulated during hypoxia? One plausible possibility is that hypoxia-specific transcription factors or epigenetic machineries are recruited to their promoter or cis-regulatory sites to induce their activation. If this is the case, expression of the different isoforms per se is completely irrelevant as the important end point is the accumulation of the protein.”

We apologize again for not being clear. Many studies have demonstrated the functional importance of NODAL, NANOG and SNAIL in stress-induced cancer stem cell phenotypes and we corroborate these findings in Figure 1. Our results explain at least in part how the levels of the latter proteins are maintained or induced under hypoxia and/or mTORi/paclitaxel treatments, when the synthesis of most proteins is inhibited. We show that the protein levels of NODAL, NANOG and SNAIL increase or are maintained under hypoxia, despite reductions in corresponding total transcript levels (Fig 1). Please note, that total mRNA levels were quantified using CDS-specific primers and thus represent sum of the expression of all mRNA isoforms for a given gene. Based

on this, the effects of hypoxia and drug treatments on NODAL, NANOG and SNAIL protein levels are uncoupled from the effects on total mRNA levels and as we demonstrate in the revised version of the manuscript correspond to changes in polysome association of selected isoforms, which confirms their translational regulation (Figures 2 and 3). To this end, the biological relevance of our findings is that selective translation of NODAL, NANOG and SNAIL mRNA isoforms sustains or increases their levels even when their corresponding mRNA levels (sum of all mRNA isoforms encoding for NODAL, NANOG or SNAIL) are reduced or unchanged, respectively. We however agree with the reviewer that the induction of specific mRNA isoforms under hypoxia may be a consequence of changes in TSS selection under hypoxia via e.g. epigenetic mechanisms. We discuss this in the revised manuscript. Identifying potential epigenetic factors implicated in TSS switching (which may give rise to different 5'UTR isoforms) is an active line of future investigation in our labs, but we trust it is out of the scope of the present study.

“2. The authors demonstrate in Figures 1-2 a correlation between upregulation of specific 5'UTR isoforms of NODAL, NANOG, and SNAIL and downregulation of mTOR during hypoxia, leading to an increase in CSC/EMT signatures. Figure 3 delves into functional outcomes of inhibiting mTOR in breast cancer cell lines in vitro and in vivo, however, the authors do not query the expression of NODAL, NANOG, or SNAIL nor their isoforms in these later assays. Why do the authors solely focus on the effects of different inhibitors/chemotherapy agents in the later figures without correlating them to the effects on the translation of these three factors? There seems to be a critical logical gap in the transition between Figures 1-2 and the remainder of the figures, and a discrepancy in the data presented with the manuscript title.”

According to reviewer's suggestions, we performed polysome profiling in cells treated with mTOR inhibitors which showed selective translation of NODAL, NANOG, and SNAIL mRNA isoforms which mirrored results obtained under hypoxia. These results are presented in Figure 3.

“3. It is not clear why SNAIL transcripts are downregulated in SUM149 but upregulated in T47D after cells being in hypoxia for 6h, yet in both cells more SNAIL protein is accumulated.”

It is likely that the different cell lines employ different mechanisms of regulation at the transcript level. What is stark, is that protein levels increase in both cell lines more than what we observe at the mRNA level. We attribute this to the common activation of selective translation. Moreover, we comment on different dynamics of the changes in mRNA and protein levels between the cell lines in the revised version of the text.

“4. What is the effect of depleting NODAL by shRNA in cells under hypoxia?”

We have previously determined that depleting NODAL with shRNA reduces hypoxia-associated phenotypes such as cellular invasion in breast cancer cells⁴. Consistent with these findings, in Figure 1, we demonstrate that depleting NODAL by shRNA or inhibiting NODAL signaling with SB341542 similarly mitigates hypoxia-associated sphere formation (Figure 1c,d).

“5. The authors need to discuss why the PDX and the SUM149-derived xenografts show opposite results in the INK condition (Fig. 3h and 4j).”

We do not suggest opposite results. One model (PDX401) does not respond to mTORi (INK128), and the other model (SUM149) initially does. This may be explained by the different requirement for mTOR signaling between these two models. Notably, the PDX has never been grown in culture, whereas SUM149 is a cell line, adapted to growth conditions which activate mTOR, and hence the latter model may be more dependent on mTOR signaling. Regardless, we believe, that these models reflect heterogeneity of patient populations, which respond differently to mTOR targeted therapies (e.g. ⁵) and highlight that despite these differences, they both respond to mTOR inhibition by increasing the percentage of BCSC-like cells.

“6. Fig. 1h is not referred in the text. Fig. 3d: bar graph and FACS profiles do not match.”

Fig. 1h has been referred to as follows: “Finally, SB431542 treatment abrogated the hypoxic induction of sphere formation in T47D and SUM149 cells, further corroborating that NODAL facilitates plasticity in response to hypoxia (**Figure 1h; Supplemental Figure 1f**)”. We have edited Fig. 3d (now 4d) accordingly.

“7. Treatment of ER+ breast cancer and TNBC patients is radically different. Along the manuscript experiments are performed using ER+ and TNBC cells without a proper justification and rationale. The authors should clearly indicate why some experiments were done with two different breast cancer subtypes and others only with one subtype.”

While ER+ and TNBC patients may in some cases be treated differently, the mechanisms governing plasticity and stem cell associated phenotypes appear to remain rather similar. We used both models *in vitro* but we opted for TNBC models *in vivo*, so that we could avoid using estrogen supplementation in the latter assays (which we worry may create a confounding variable due to its documented roles in the regulation of translation)⁶. We have added this explanation into the manuscript in the Methods related to animal modelling.

“8. Authors should be more specific when describing results, i.e., quantitative numbers should be included in parentheses where applicable.”

We have corrected the manuscript accordingly.

“9. Representative pictures of spheres throughout the manuscript are of one sphere to highlight difference in sphere sizes between treatment conditions, however, sphere-forming ability was demonstrated using quantifications of sphere number. Were sphere sizes also quantified? How was sphere counting performed? Specifically, was there a cutoff sphere size that was considered for counting? The methods are not clearly detailed in the text nor the methods section.”

We have updated the methods to more clearly indicate key aspects of these experiments including choosing a dilution that ensures fewer than one sphere per well, the size cut off used and how we counted. In general, we did not quantify sphere size. Where we did, we have updated the methods. We have included more detailed description of experimental procedure and we also provide quantification of sphere sizes in Supplemental Figure 5e, where notable differences in size were observed.

“10. Figures 3 and S3 are dedicated to demonstrating that mTOR inhibition increases the CSC/EMT signature of breast cancer cells.”

“a. In line with comment #7, all in vitro work was performed in non-highly metastatic cell lines T47D and MCF7. However, in vivo xenografts use the highly metastatic SUM149 line. The authors could elaborate, again, on the choice of cell lines for each assay.”

Please see response to comment 7. Also, *in vitro* work has also been conducted with SUM149 cells and in some cases MDA-MB-231.

“b. Methods state that 700ul total volume (containing 5x10⁵ cells) were injected into the tail veins of mice – more than double the maximum volume according to NIH guidelines. Is this a typo error?”

Thank you for pointing this typographical error out. We have corrected this to read 100 μ L.

“c. The focus of this figure is to demonstrate the increase in CSC/EMT signatures upon treatment with INK, i.e., mTOR inhibition. INK inhibition in a PDX model also increases primary tumor growth (Figure 3h). The authors do not perform additional assays characterizing the primary tumors (only a sphere assay, Figure 3i), nor do they discuss in better detail how this result relates to their prior observations. Additionally, limiting dilutions xenografts would be better suited to demonstrate CSC frequencies in vivo after INK inhibition.”

We have revised the manuscript to indicate how the results relate to prior observations. Essentially, the intention was to demonstrate that similar phenotypes are induced *in vivo* as are induced *in vitro*. In place of limiting dilution assays, we have included a lung colonization assay as a surrogate. This is now clearly stated. As an important control, we have measured CA9 (for hypoxia), pphosho and total 4E-BP1 levels in the tumours. These results are included in Supplemental Figure 4 and demonstrate that INK is on target. Importantly, these data also demonstrate that pphosho-4E-BP1 levels are inversely correlated with CA9 staining, which shows that mTOR is suppressed in hypoxic regions of the tumors *in vivo* (as seen *in vitro* in Figure 1).

Reviewer #2 (Remarks to the Author):

“In this manuscript the authors describe the selective translation of isoforms of NANOG, SNAIL and NODAL in breast cancer cells exposed to hypoxia. These findings suggest a model

of translational regulation whereby differences in 5'UTR features of the mRNA isoforms harboring the same ORF allow isoform-specific translation and consequent upregulation of NANOG, SNAIL and NODAL proteins under hypoxia. This is considered by the authors as the induction of an adaptive stem cell program which leads to the acquisition of breast cancer cells with stem cell-like features (BCSC) phenotypes. This process was also induced by mTOR inhibitors and chemotherapeutics and was inhibited by an Integrated Stress Response Inhibitor (ISRIB). Although this is an interesting manuscript several points remain stratification.”

We would like to thank the reviewer for finding our manuscript interesting.

“Fig. 1h: control SB normoxia is missing and Fig. 3b,c,d: SB alone experiments are missing.”

We agree with the reviewer and have incorporated appropriate control in Figure 1h. This is the same control as would be used in Figure 4 (formerly 3). We find that SB431542 on its own does not affect sphere formation in T47D cells (Figure 1h) and that it slightly reduces it in SUM149 cells (Supplemental Figure 1f). This difference may reflect the presence of endogenous TGF β /NODAL/ACTIVIN signaling in SUM149, but not T47D, cells.

“Fig. 1i: To better appreciate the exact role of hypoxia, adjusted normoxic controls, (f.e. 12h, 24h) are needed and Fig. 4b: To better appreciate the exact role of AZE, adjusted controls, (f.e. 12h, 24h) are needed.”

We accounted for this important potential confounder in our experimental design: Time in culture was controlled for by placing cells in hypoxia (or starting AZE treatment) at different times so that all samples were analyzed the same time after seeding. For example, 0h cells were cultured for 24h in normoxia, 6h hypoxic cells were cultured for 18h in normoxia followed by 6h in hypoxia and 24h hypoxic cells would be cultured for 24 h in hypoxia. We have now clarified this in the methods.

“Fig 1k: Statistical analyses are missing. How many independent replicates were performed? And Fig. 4a: Statistical analyses are missing. How many independent replicates were performed?”

We have included statistical analysis for all ‘Global Translation’ graphs (Figure 1k, Supplemental Figure 1i,j). These experiments were performed in 3 independent biological replicates. This information is provided in the corresponding figure legends. We hope that this helps increase confidence in the global suppression of translation in response to hypoxia. Likewise, we have included a new experiment in Figure 3 that demonstrates a similar phenomenon in response to chemically induced translation inhibition. In this case all replicates are included in Supplemental Figure 3. Densitometric analyses of Western blots (n=3) from Figure 5a are provided in Supplementary Figure 5a. This figure also contains the Western blot analysis and quantification of additional 3 replicates of the same experiment performed in SUM149 cells.

“Fig 1n: Only one loading control (actin) is displayed. This most probably does not account for all WBs displayed. Please show appropriate loading controls for all experiments as well as quantification.”

We have included quantification of at least 2 independent replicates. The densitometric analyses of the Western blots are included in the Supplemental Figure 1k. Note that the loading controls, in most cases, are the total protein (corresponding to the phosphorylated version). This has now been clearly articulated.

“Fig. 3f: Only one loading control (actin) is displayed. This most probably does not account for all WBs displayed. Please show appropriate loading controls for all experiments as well as quantification.”

We have included additional loading controls as indicated in the revised text. We have included quantification of 3 independent replicates. The densitometric analyses of the Western blots are included in the Supplemental Figure 4i. We also included western blot analyses and quantification of 3 independent replicates of the same experiment in SUM149 cells.

“Fig. 3a: Quantification is missing”

We have included quantification of 3 independent replicates. The densitometric analyses of the western blots are included in the Supplemental Figure 4a. We also included western blot analyses and quantification of 3 independent replicates of the same experiment in MCF7 cells.

“Fig. 5e: If sphere size has been measured this does not seem to be increased by hypoxia. Please specify approach.”

The sphere size has been quantified for Figure 5e and this analysis can be found in Supplemental Figure 5e. In this model, hypoxia did not seem to increase sphere size; however the inability to evoke the ISR seemed to cause a significant reduction in size, that was exasperated in hypoxia. We have updated the methods to more clearly indicate key aspects of the experiment including a description of how dilutions were chosen to ensure fewer than one sphere per well; the size cut off used; and how we counted the spheres. We think that investigating sphere size is valuable and intertwines with complicated questions with which we are actively pursuing such as the role of cellular energetics and/or cell size (with mTOR at its core) in the cycling of cancers between growth, dormancy, adaptation and regrowth.

“Fig. 4g, h: Only one loading control (actin) is displayed. This most probably does not account

for all WBs displayed. Please show appropriate loading controls for all experiments as well as quantification.”

As above, appropriate controls were included in Figure 5h and we performed requested quantification experiments, now included in Supplemental Figure 5f, g.

“The quality of Nanog and Sox2 WB is poor. Solvent controls are missing. Fig. 4h: Correct typo with Salubrinal.”

We have improved the quality of NANOG Western blots, included the appropriate controls (Supplementary Figure 5h) demonstrating that 0.1% v/v DMSO does not increase the expression of NANOG or SNAIL and corrected the typo. Note that we chose to remove SOX2 as our Western blots were not of optimal quality and we did not feel that this added to the already complex data within this paper.

“Fig. 5a,b: Only one loading control (actin) is displayed. This most probably does not account for all WBs displayed. Please show appropriate loading controls for all experiments as well as quantification. The quality of Sox2 WB is poor. Solvent controls are missing.”

As above, these issues were rectified in the revised version of the manuscript. We have added solvent controls demonstrating that 0.1% v/v DMSO does not induce any transitional inhibition via phosphorylation of eIF2 α or 4E-BP1 (Supplementary Figure 5b), nor does it increase the expression of NANOG or SNAIL (Supplementary Figure 5h) Loading controls for phosphorylated proteins were their matched total protein as described in the legends. Proteins used as loading controls are now indicated in the legends for the densitometry graphs (Supplementary Figure 6a,b).

“Fig. 5g: How exactly were viable cells determined? Histology suggests a difference between ISRIB and Pac which is not displayed in the graph.”

We have selected new sections to better represent the mean data of each group. Though, as the reviewer rightly noted, there may be differences between the means of paclitaxel and ISRIB that fall below statistical significance due to the heterogeneous response within the paclitaxel treatment group. We comment on this and have also included a more detailed set of methods to more clearly illustrate how viable cells were identified as outlined below:

“All IHC images were analyzed using ImageJ. Total tumor section area was selected and measured by thresholding against background pixel intensity. Necrotic regions are comprised of pixels of higher uniform intensity compared to the intact tumour cells and a second round of thresholding against the lighter necrotic regions was used to distinguish and measure the total area of necrosis. Viable cells in hypoxic regions were determined by identifying dense, dark areas within necrotic regions. These dark foci represent cells with intact nuclei and the area occupied by these cells was divided by the area of the necrotic region to capture percent of viable cells in hypoxia regions.”

“Suppl Fig. 1j: No loading controls are displayed. No controls for 12h, 24h values are shown. This is also true for many subsequent figures in the supplement.”

As above, these issues were rectified in the revised version of the manuscript. For many of the translational regulators, the loading control was the unphosphorylated protein. As described above, our experimental design accounted for confounders associated with time in culture.

“Overall, the number of independent replicates is not clear throughout the study which raises concerns regarding reproducibility and validity.”

We apologize profoundly for this omission. We have clearly indicated number of independent replicates for each experiment in figure legends. All the experiments were carried out in at least 2-3 independent replicates as indicated. We have also added the requested quantifications to the manuscript to improve statistical confidence and demonstrate biological variability.

“In addition, although the use of patient derived xenografts is of interest, conclusions are drawn from one well-differentiated versus one poorly differentiated patient sample. More set-ups from different patients are needed to support the conclusions.”

We performed our *in vivo* experiments using two different cell line derived xenograft (PDX) models and two patient derived xenograft models. While we agree that additional PDX models would be needed to better approximate the efficacy of ISRIB in breast cancer patients, for example, we feel that the use of four models, sufficiently demonstrated that mTOR inhibitors and stress can induce BCSC-phenotypes and that this can be mitigated by ISRIB.

“Furthermore it would be of interest to see whether the different gene isoforms are indeed present in different breast cancer patient populations and which of them relates to chemoresistance.”

We agree that this would be of interest. Unfortunately, the large data sets required to perform this type of analysis with accuracy do not annotate all of the isoforms; precluding our ability to conduct this work with available resources. This is compounded by the fact that total mRNA levels are discordant with protein levels and that we cannot decipher the isoforms at the protein level as they all encode the same proteoform. While out of the scope of the current study, we intend to analyze RNA from breast cancer patients (with outcome information) using isoform-specific technologies.

“A scheme summarizing the main findings would be beneficial since the manuscript is at certain points lengthy and not very focused.”

We included schematic presentations of the model (Figure 3 and 7).

Reviewer #3 (Remarks to the Author):

“This very interesting manuscript by Jewer et al. describes a mechanism of translational plasticity underlying breast tumor progression and its therapeutic responses. Specifically, in the first part of the manuscript, the authors report that breast cancer cells translate specific mRNA isoforms of NANOG, SNAIL and NODAL transcripts under hypoxic conditions. The authors further showed that mTOR inhibition as well as chemotherapeutic agents induce translation activation of NANOG, SNAIL and NODAL mRNAs, which in turn cause breast cancer cells to acquire stem cell-like phenotypes, ultimately contributing to tumor growth. Interestingly, this translation reprogramming downstream these stress agents are functionally linked to eIF2 α phosphorylation (P-eIF2 α). Importantly, ISRIB, which inhibits the activity of P-eIF2 α , represses tumor growth or the acquisition of stem-cell phenotypes in xenografts or PDX models treated with mTOR inhibitor treatments or paclitaxel.

This is an important paper that highlights the interplay between distinct pathways that converge on manipulating the translational machinery as a mechanism to drive and maintain tumor growth even in the presence of anti-cancer drugs. The manuscript is timely and deserves to be published; however, I have a few comments that should be addressed before publication.”

We would like to thank the reviewer for finding our study important.

“-To tie the first part of the paper with the second, the authors should analyze the RNA level as well as the translation efficiency of the specific isoforms (NANOG, SNAIL and NODAL mRNAs) after mTOR inhibition, chemotherapeutics, and with or without ISRIB as they performed in Figure 2 upon hypoxia.”

We have performed polysome profiling experiments with mTORi (INK128) which largely mirrored findings obtained in hypoxia. These data are included in Figure 3. Interestingly, we determined that the selective increase in translation of a subset of isoforms encoding stemness factors during mTOR inhibition can be mitigated by ISRIB. This was despite the inability of ISRIB to restore global translation upon mTOR inhibition as described previously⁷. This suggests that the ISR is a major driver of plasticity induced as a consequence of mTOR inhibition and that it does not mechanistically account for the reduced protein synthesis that occurs with these treatments.

“- There are several reports in literature showing that mTOR activation actually results in increased transcription and translation of EMT signatures, including SNAIL, Vimentin and NANOG. Therefore, the authors should clarify (in the discussion section) that the effect they observe on the translation upregulation of specific mRNAs with mTOR inhibition is very specific to these breast cancer cell types and treatment conditions. This is important as while the authors interestingly found that these mRNAs possess different isoforms with a distinct 5'UTR, it is unclear which of these isoforms will eventually contribute to protein abundance in the cells. For example, can the authors mutate the regulatory uORF in the Nodal mRNAs and assess the protein level upon normoxia and hypoxia?”

We would like to thank the reviewer for raising this important point. To this end, we modified the discussion section to reflect her/his concerns. Indeed, we suggest a bimodal role for mTOR in tumour growth and progression. In this model mTOR is needed for growth and likely for the sustained production of EMT and stem cell factors, once plasticity has been evoked. Indeed, once proteins such as NODAL are induced, they are able to reprogram the epigenome and to even sustain their own expression through unrestricted positive feedback loops^{4, 8}. However, we propose that mTOR inhibition (concomitant with the ISR) enables plasticity. This occurs by the selective translation of stem cell associated factors, which may not be sufficiently abundant relative to other proteins in a rapidly growing cell, but which can accumulate and drive plasticity during stress. This is a hypothesis first proposed by Adami, over 100 years ago, and also likely explains phenomena such as senescence-associated plasticity. While out of the scope of this study, we are actively pursuing the role of energetic switching (with mTOR at its core) in the cycling of cancers between growth, dormancy, adaptation and regrowth.

We have performed polysome profiling in conjunction with isoform specific RT-qPCR, which revealed isoforms that contribute to protein levels (Figures 2 and 3). We confirmed this with reporter assays (Figure 2g-k) which demonstrated that only specific isoforms are well translated in hypoxia. Using primers common to all isoforms (Figure 1m) we demonstrated that NANOG, SNAIL and NODAL are selectively translated in hypoxia. Also, due to the shared sequence, the smaller isoforms of NODAL and SNAIL measured in the polysome profiles in Figures 2 and 3 necessarily included the larger isoforms as well. These total values represent the sum of all isoforms, suggesting that those which are selectively translated must be contributing significantly to protein synthesis. The functionality of uORF in *NODAL* 298 mRNA was validated using reporter assays which served as a surrogate for protein levels (Figure 2k). Interestingly, not all of the isoforms which were well translated contain a uORF, suggesting numerous potential modes of regulation which we shall explore in future studies.

“-In Figure 4, panel C should be quantified as it is not clear what the ATF levels are under different conditions.”

These quantifications have been performed and are included in the revised version of the paper in Supplemental Figure 5d.

“- It would be interesting if the authors can assess (if it is possible) whether the increase of P-PERK and P-eIF2 α can predict for poor survival in breast cancer patients.”

Considering the past issues with the use of phospho-eIF2 α Ab in tissues, we were not confident enough to use publicly available data that are also sparse and not well-curated when it comes to phospho-eIF2 α staining. We are therefore initiating a collaboration with Le Quesne (University of Leicester) who overcame the issues with phospho-eIF2 α staining in tumor tissues to address this very valid point. We are however still in the process of collecting breast cancer samples to obtain

a well-curated tissue bank of the size which will be appropriate for such a study. Of note, Le Quesne recently completed a study with one of our coauthors (Dr. Koromilas (LDI/McGill)) which shows that high phospho-eIF2 α levels correlate with poor prognosis in lung cancers. This study is submitted for publication and in discretion, we can show the pertinent data to the reviewer.

REFERENCES

1. Quail, D.F., Taylor, M.J. & Postovit, L.M. Microenvironmental regulation of cancer stem cell phenotypes. *Curr.Stem Cell Res.Ther.* **7**, 197-216 (2012).
2. Quail, D.F., Siegers, G.M., Jewer, M. & Postovit, L.M. Nodal signalling in embryogenesis and tumorigenesis. *Int.J.Biochem.Cell Biol.* (2013).
3. Hepburn, A.C. *et al.* The induction of core pluripotency master regulators in cancers defines poor clinical outcomes and treatment resistance. *Oncogene* **38**, 4412-4424 (2019).
4. Quail, D.F. *et al.* Low oxygen levels induce the expression of the embryonic morphogen Nodal. *Mol.Biol.Cell* **22**, 4809-4821 (2011).
5. Janku, F. *et al.* PI3K/AKT/mTOR inhibitors in patients with breast and gynecologic malignancies harboring PIK3CA mutations. *Journal of clinical oncology : official journal of the American Society of Clinical Oncology* **30**, 777-782 (2012).
6. Lorent, J. *et al.* Translational offsetting as a mode of estrogen receptor alpha-dependent regulation of gene expression. *Embo j* **38**, e101323 (2019).
7. Sidrauski, C. *et al.* Pharmacological brake-release of mRNA translation enhances cognitive memory. *eLife* **2**, e00498 (2013).
8. Hendrix, M.J. *et al.* Reprogramming metastatic tumour cells with embryonic microenvironments. *Nat.Rev.Cancer.* **7**, 246-255 (2007).

REVIEWERS' COMMENTS:

Reviewer #1 (Remarks to the Author):

The authors made a great effort in clarifying my comments. I support publication of this very nice study.

Reviewer #2 (Remarks to the Author):

In this manuscript the authors aim to link the occurrence of multiple isoforms of NANOG, SNAIL and NODAL transcripts in breast cancer cells and the specific induction of translation of a subset of them by hypoxia to induction of breast cancer plasticity and "fate-switching" towards stem-cell like phenotypes. Although this is an interesting approach several points remain unclear.

Fig. 3: None of the data displayed seems to be of statistical significance. Why then display this figure?

Although hypoxia and mTOR are linked the link between mTOR inhibitors and hypoxia in the context of translational regulation and plasticity is not well elaborated at the experimental level. This would add to the coherence of the study.

Although a PDX line has been employed the study unfortunately does not add patient data which would be important to give an idea about the overall clinical importance of the complex findings. The summary figure is not really clear and does not help too much in integrating all complex aspects of the manuscript.

The manuscript is rather lengthy and could be better focused on more rigorously explaining the data.

Minor: There are some minor grammar/spelling mistakes throughout the manuscript.

Reviewer #3 (Remarks to the Author):

The authors satisfactorily addressed all of my comments. This very interesting paper is suitable for publication.

We would like to thank you and the reviewers for the insightful comments and the opportunity to revise our manuscript. To this end, we have substantially expanded our studies and clarified explanations regarding the role of differential isoform translation under hypoxia, its impact on protein levels and functional significance. Moreover, we now provide direct evidence corroborating differential sensitivity of isoform translation to mTOR inhibition. Finally, we addressed the comments pertinent to potential clinical relevance of isoforms.

Reviewer #1 (Remarks to the Author):

The authors made a great effort in clarifying my comments. I support publication of this very nice study.

Reviewer #2 (Remarks to the Author):

In this manuscript the authors aim to link the occurrence of multiple isoforms of NANOG, SNAIL and NODAL transcripts in breast cancer cells and the specific induction of translation of a subset of them by hypoxia to induction of breast cancer plasticity and “fate-switching” towards stem-cell like phenotypes. Although this is an interesting approach several points remain unclear.

Fig. 3: None of the data displayed seems to be of statistical significance. Why then display this figure?

Upon the reviews request we have included the statistical tests for this figure.

Although hypoxia and mTOR are linked the link between mTOR inhibitors and hypoxia in the context of translational regulation and plasticity is not well elaborated at the experimental level. This would add to the coherence of the study.

We have now included multiple descriptions of the mTOR/4E-BP1 and ISR/eIF2a signalling that occurs in hypoxia and how it is mirrored by mTOR inhibition and chemotherapeutic stress. We hope this helps clarify the convergent stress signalling of these three conditions.

Although a PDX line has been employed the study unfortunately does not add patient data which would be important to give an idea about the overall clinical importance of the complex findings.

We acknowledge the importance of patient data, to that end we have included patient data from the TCGA demonstrating that lower expression of translation inhibiting proteins 4E-BP1 and PERK are associated with better survival and that these proteins are elevated in tumours relative to normal adjacent tissues.

The summary figure is not really clear and does not help too much in integrating all complex aspects of the manuscript.

We have added conceptual descriptions of the translational landscape which we believe helps to clarify the conclusions without the need to focus on the specific isoforms.

The manuscript is rather lengthy and could be better focused on more rigorously explaining the data.

We have appreciated this feed back and have cut down the length significantly while clarifying the explanations of our data.

Minor: There are some minor grammar/spelling mistakes throughout the manuscript.

Reviewer #3 (Remarks to the Author):

The authors satisfactorily addressed all of my comments. This very interesting paper is suitable for publication.